# MQuant: Unleashing the Inference Potential of Multimodal Large Language Models via Full Static Quantization

## Abstract

Recently, multimodal large language models (MLLMs) have garnered widespread attention due to their ability to perceive and understand multimodal signals. However, their large parameter sizes and substantial computational demands severely hinder their practical deployment and application. While quantization is an effective way to reduce model size and inference latency, its application to MLLMs remains underexplored. In this paper, we conduct an in-depth analysis of MLLMs quantization and identify several challenges: slow inference speed of the visual tokens, distributional differences across modalities, and visual outlier clipping degrades performance. To address these challenges, we propose *MQuant*, a quantization framework tailored for MLLMs. Specifically, 1) we design Modality-specific Quantization (MSQ) and Attention-Invariant Flexible Switching (AIFS) to support per-tensor static quantization and facilitate efficient inference. 2) we introduce an unified LayerNorm-to-RMSNorm transformation, achieving seamless integration of the MLLM vision encoder with Hadamard rotation. 3) we propose Rotation Magnitude Suppression (RMS) to mitigate outliers introduced by Hadamard rotation. Experiments conducted on five mainstream MLLMs demonstrate the superior performance and broad applicability of *MQuant*. For example, it maintains around 98% of the floating-point accuracy under the W4A8 setting. To the best of our knowledge, *MQuant* is the first quantization solution for MLLMs, paving the way for future advancements in their application.

## 1 Introduction

Recently, large language models (LLMs) such as GPT (Brown et al., 2020) and the LLaMA series (Touvron et al., 2023a;b; Dubey et al., 2024) have shown remarkable success across various domains, but they often struggle when handling non-textual data. Multimodal large language models (MLLMs) (Reid et al., 2024; Achiam et al., 2023b; Wang et al., 2023) overcome this by incorporating various modalities such as images and video, enabling a comprehensive understanding of diverse data types. However, the vast number of parameters and substantial computational demands due to the processing of multimodal inputs, which severely hinders their practical application. This is especially true in resource-constrained and privacy-sensitive environments.

Quantization is an effective technique for compressing and accelerating neural networks by converting high-precision values (e.g., FP32) to low-precision values (e.g., INT8), significantly reduces memory usage and inference latency. While quantization has been widely applied to LLMs (Yuan et al., 2023a; Shao et al., 2023), its application to MLLMs is essential but remains relatively under-explored.

When quantizing MLLMs, it is essential to consider the differences in inference efficiency that arise from their multi-modal nature. A typical difference is the Time to First Token (TTFT) in MLLMs increases significantly with the resolution and aspect ratio of input images or videos. As shown in Fig 1(a), in models such as Qwen2-VL (Wang et al., 2024), the number of prefill visual tokens grows as image resolution increases (detailed figure in Fig. 7). This rapid expansion in token count exacerbates the inference latency, particularly in per-token dynamic quantization, which requires separate processing for each token, such as memory access and scale computation. As a result, the TTFT in MLLMs increases drastically, severely impacting overall inference latency. Moreover, in higher-demand scenarios such as video-based tasks and multi-image dialogues, the accumulation of visual tokens becomes even more pronounced, further exacerbating the increase in TTFT.

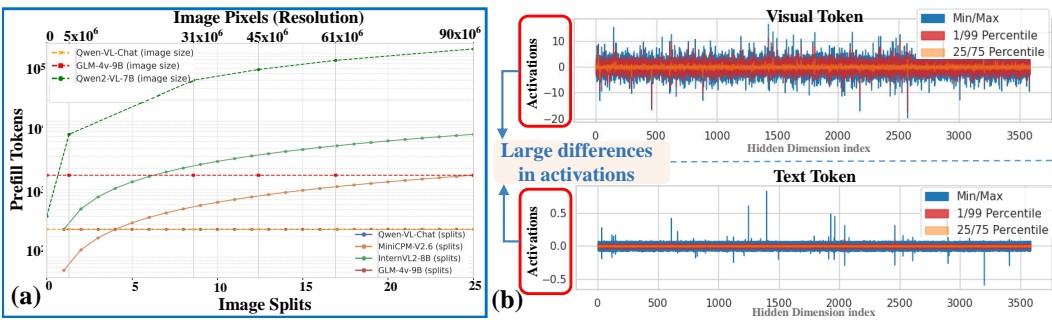

Figure 1: (a) The number of prefill visual tokens across different MLLMs as the image splits or resolution increases. (b) The activation values of visual tokens range from $-20$ to $10$, whereas textual tokens are centered around $0$, with only a few having absolute magnitudes exceeding $0.5$.

One intuitive approach to address this issue is to employ per-tensor static quantization for prefill multimodal tokens, which eliminates the need for token-wise scale computation. While this approach achieves up to 66% acceleration across all LLM linear layers (Fig. 4), it still results in a serious accuracy degradation. We investigate the root causes and identify the following key challenges in MLLM quantization:❶ **Balancing efficiency and accuracy when adopting per-tensor static quantization for multi-modal tokens.** In MLLMs, input multi-modal tokens include visual and textual with distinct value distributions. Unlike per-token dynamic quantization, which can adaptively compute token-wise scales in an online manner, static quantization applies few scale factors across the mixed multi-modal tokens in an offline manner, leading to a difficult trade-off between efficiency and accuracy. ❷ **The data distribution of visual and textual tokens exhibits significant differences.** Unlike LLMs, which solely handle textual features, MLLMs face the challenge of processing input with distinct distributions across multiple modalities. As shown in Fig 1 (b), the activation distributions between visual and textual tokens reveal substantial numerical discrepancies. Joint quantization of these tokens may cause the larger values of visual tokens to overshadow the smaller textual values, resulting in a loss of textual information. Therefore, MLLMs require modality-specific approaches to account for the heterogeneous nature of visual and textual token distributions.

❸ **Clipping outliers in visual tokens leads to performance degradation.** As shown in the Tab. 1, applying a stricter clipping range causes a large drop in accuracy for visual tokens, while textual tokens exhibit only a minor decrease. In contrast, using a full clipping range slightly mitigates the performance drop for visual tokens. This indicates that careful handling of outliers in visual tokens is crucial—indiscriminate clipping of outliers leads to severe accuracy loss.

Table 1: Results of the Qwen-VL on TextVQA using different clipping ranges for visual and textual tokens. Activations are quantized using per-tensor static setting.

| Clip Range | Bits (W/A) | Visual Tokens | Textual Tokens |
|---|---|---|---|
| - | BF16 / BF16 | 61.40 | |
| (0-0.99999) | BF16 / INT16 | **18.92** (↓**42.48**) | 60.09 (↓1.31) |
| (0-1.0) | BF16 / INT16 | 61.2 (↓0.20) | 61.25 (↓0.15) |

To this end, we propose *MQuant*, an accurate and efficient post-training quantization (PTQ) solution specifically designed for multimodal large language models (MLLMs). First, to reduce the TTFT while maintaining accuracy, we introduce **Modality-Specific Quantization (MSQ)**, which handles the distinct distribution differences between visual and textual tokens. For further acceleration, we design an **Attention-Invariant Flexible Switching (AIFS)** scheme. AIFS transforms mixed multimodal tokens into unified, modality-decoupled tokens without the need for dynamic position vectors, maintaining computational equivalence and memory efficiency by avoiding the overhead associated with dynamic processing. Second, based on the *computation invariance* (Ashkboos et al., 2024a), we further develop a **Post-LN + Rotate** scheme to extend and accommodate various MLLMs architecture, enabling seamless integration with rotation to smooth outliers. Third, we reveal the weight outliers caused by the online Hadamard rotations through theoretical analysis and propose **Rotation Magnitude Suppression (RMS)** to mitigate them, further improving quantization accuracy.

We evaluate *MQuant* on various datasets and five mainstream MLLMs, including InternVL (Chen et al., 2024a), Qwen-VL (Bai et al., 2023b), MiniCPM-V (Yao et al., 2024), CogVLM2 (Hong et al., 2024) and Qwen2-VL (Wang et al., 2024). The results demonstrate that *MQuant* achieves less than 1% accuracy loss on all MLLMs under the W4A8 setting, highlighting its superior performance and wide applicability. We publish the code, hoping to provide new insights and advance the deployment and application of MLLMs. Our main contributions are summarized as follows:

- We present the first comprehensive analysis of quantization issues in MLLMs, unveiling the root causes of performance collapse and identifying inference speed bottlenecks as well as the quantization challenges posed by modality differences.
- We design Modality-specific Quantization (MSQ) and Attention-Invariant Flexible Switching (AIFS) to support static quantization. Besides, we propose a Post-LN + Rotate scheme tailored to accommodate various MLLMs.
- We analyze the root issues of weight outliers caused by online rotation, and propose Rotation Magnitude Suppression (RMS) to effectively mitigate it.
- We propose *MQuant*, to the best of our knowledge, the first PTQ solution for MLLMs.

## 2 RELATED WORK

**Multimodal Large Language Models.**  The rapid advancement of LLMs has spurred significant progress in MLLMs. Flamingo (Alayrac et al., 2022) pioneered connecting pre-trained visual encoders to LLMs, demonstrating strong generalization across visual-language tasks. Following the emergence of ChatGPT Achiam et al. (2023a), numerous open-source models based on pre-trained LLMs like LLaMA (Touvron et al., 2023a) and its variants (Touvron et al., 2023b; Zheng et al., 2024) have been proposed (Li et al., 2023; Huang et al., 2024; Zhu et al., 2023; Liu et al., 2024). Subsequent efforts like Qwen-VL (Bai et al., 2023b), InternVL (Chen et al., 2024b), and CogVLMV2 Hong et al. (2024) enhanced MLLMs from perspectives such as high-resolution input and larger-scale training data. However, the substantial parameters of MLLMs lead to high computational costs, limiting broader application. Recently, smaller MLLMs like Mini-Gemini (Li et al., 2024), MobileVLM Chu et al. (2023; 2024), and MiniCPM-V (Yao et al., 2024) have emerged. Despite these developments, dedicated quantization methods for MLLMs to further reduce memory usage and accelerate inference remain under-explored.

**Post-Training Quantization for LLMs.**  Existing post-training quantization (PTQ) methods for LLMs are categorized into weight-only and weight-activation quantization (Zhao et al., 2023; Yuan et al., 2024). Weight-only methods like GPTQ (Frantar et al., 2022), QuIP (Chee et al., 2024), and AWQ (Lin et al., 2023) achieve high compression rates but offer limited inference acceleration. In contrast, weight-activation quantization methods (Xiao et al., 2022; Wei et al., 2022; Yuan et al., 2023b; Zhang et al., 2024) quantize both weights and activations, improving memory usage and latency. The main challenge is activation outliers causing quantization errors. Techniques like SmoothQuant (Xiao et al., 2022) shift quantization difficulty from activations to weights, while OmniQuant (Shao et al., 2023) optimizes performance by training quantization parameters. SliceGPT (Ashkboos et al., 2024a) reduces memory demands by designing a Pre-LN + Rotate Scheme for LLMs sparsification based on computational invariance. They achieve this by adding a linear layer in the residual connection (see Appendix A.14). Unlike SliceGPT, we further develop a Post-LN + Rotate scheme to accommodate more vision encoder and extends its applicability to various MLLMs. This enhancement broadens the the LayerNorm + Rotate approach, making it suitable for both Pre-LN and Post-LN configurations across various MLLMs. Recently, Quarot (Ashkboos et al., 2024b) introduces rotations to eliminate outliers; however, this solution is not applicable to MLLMs due to inherent modality differences.

## 3 PRILIMINARY

### 3.1 MLLM ARCHITECTURE

The existing MLLM framework  (Bai et al., 2023b; Chen et al., 2024b) consists of three primary modules: a vision encoder $\mathbf{E}$ for processing visual inputs and feature extraction, a visual-language projector $\mathbf{P}$ that serves as a bridge to align the two modalities and a large language model ($LLM$) that handles the multi-modal tokens and performs reasoning.

**Vision Encoder**   Taking the input image or video $\boldsymbol{X}_v$ as input, Existing vision encoders typically adhere to the Vision Transformer (ViT)  (Dosovitskiy, 2021) architecture, such as CLIP (Radford et al., 2021), OpenCLIP (Ilharco et al., 2021) and SigLIP (Zhai et al., 2023). the vision encoder compresses the original vision information into more compact patch features $\boldsymbol{F}_v$, which can be formulated as:

$$\boldsymbol{F}_v = E(\boldsymbol{X}_v) \tag{1}$$

**Vision-Language Projector**   The task of the vision-language projector $\boldsymbol{P}$ is to map the visual patch features $\boldsymbol{F}_v$ into the textual feature space:

$$\boldsymbol{E}_v = P(\boldsymbol{F}_v) \tag{2}$$

**Large Language Model**   The pre-trained large language model serves as the core component of MLLMs, endowing the framework with exceptional capabilities, such as zero-shot generalization, instruction following, and in-context learning.

Typically, a text tokenizer is integrated with the LLM, mapping text prompts $\boldsymbol{X}_t$ to the text tokens $\boldsymbol{E}_t$. The text tokens $\boldsymbol{E}_t$ and the visual tokens $\boldsymbol{E}_v$ are then concatenated to form the input for LLMs, which outputs the final response sequence $\boldsymbol{O}_a$ in an autoregressive manner:

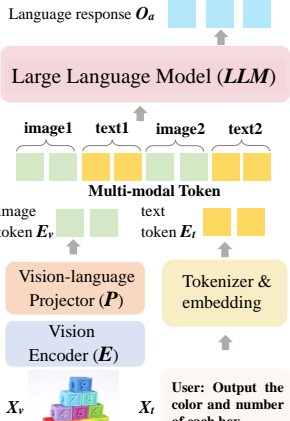

$$LLM(\boldsymbol{O}_a|\boldsymbol{E}_v, \boldsymbol{E}_t) = \prod_{i=1}^{l} LLM(y_i|\boldsymbol{E}_v, \boldsymbol{E}_t, y_{<i}) \tag{3}$$

where $l$ denotes the length of $\boldsymbol{O}_a$. The parameter sizes of large language models (LLMs) range from 3 billion to tens of billions. Commonly used open-source LLMs include the Llama series (Touvron et al., 2023a;b), Qwen (Bai et al., 2023a), InternLM (Cai et al., 2024), MiniCPM (Hu et al., 2024), ChatGLM (GLM et al., 2024).

### 3.2 Transformer and Quantization

Transformer networks (Vaswani et al., 2017) have proven highly effective across a wide range of tasks, including language modeling and visual perception. A transformer consists of multiple layers, each containing a multi-head self-attention (MHSA) block followed by a feed-forward network (FFN) block, with a pre-normalization or post-normalization layer between them. The details of whole Transformer are in Appendix A.15.

**Attention Block in MLLMs**   Given an input prompt, the generation process of LLMs can be broadly categorized into two distinct phases: the prefill phase, which computes and stores the KV cache for input tokens, and the decoding phase, where new tokens are generated through a next-token-prediction scheme. Given input data $\mathbf{X}$ after LayerNorm (LN), and an attention block with its weight matrices $\mathbf{W}_q$, $\mathbf{W}_k$ and $\mathbf{W}_v$, attention scores is $\mathbf{A}$, the prefill phase is formulated as:

$$\mathbf{Q} = \mathbf{X}\mathbf{W}_q, \quad \mathbf{K} = \mathbf{X}\mathbf{W}_k, \quad \mathbf{V} = \mathbf{X}\mathbf{W}_v \tag{4}$$

$$\mathbf{A} = \text{Softmax}\left(\frac{\mathbf{Q}\mathbf{K}^T}{\sqrt{D}}\right), \quad \mathbf{O} = \mathbf{A}\mathbf{V} \tag{5}$$

**Quantization**   Quantization maps high-precision value into discrete levels, we adopt uniform quantization (Jacob et al., 2018) in our study. Given a floating-point (FP) tensor $x$ (weights or activations), it can be uniformly quantized to $b$-bits in signed quantization as follows:

$$\hat{\mathbf{x}} = \mathbf{Q}_U(\mathbf{x}, b) = (clamp(\lfloor \frac{\mathbf{x}}{s} \rceil + z, q_{min}, q_{max}) - z) \cdot s, \quad s = \frac{\max(|\mathbf{x}|)}{2^{b-1} - 1} \tag{6}$$

where $\lfloor \cdot \rceil$ is the rounding-to-nearest operator, and the function $clamp(\cdot)$ clips values outside the integer range $[q_{min}, q_{max}]$. $z$ is zero-point. $s$ denotes the quantization scale factor, which reflects the proportional relationship between FP values and integers. $[q_{min}, q_{max}]$ is the quantization range determined by the bit-width $b$. Generally, when we quantize the network's weight with 4-bit and activations with 8-bit, called it as W4A8. We can calculate $s$ offline using the activations from calibration samples, known as **static quantization**. We can also use the runtime statistics of activations to get $s$, referred to as **dynamic quantization**. More details are in Appendix A.11.

## 4 Methodology

In this section, we present *MQuant*, a post-training quantization solution specifically designed for MLLMs. In Sec. 4.1, we describe modality-specific quantization (MSQ) and attention-invariant flexible switching (AIFS). In Sec. 4.2, we introduce the Post LayerNorm-to-RMSNorm transformation. In Sec. 4.3, we identify the weight outliers caused by the online Hadamard rotations and state Rotation Magnitude Suppression (RMS). We provide the detailed MQuant algorithm for FP MLLMs in Appendix A.5 Algorithm 1.

### 4.1 Modality-Specific Quantization and Attention-Invariant Flexible Switching

As discussed in Sec 1, increasing the resolution of input images or videos significantly raises the number of visual tokens (Fig 1 (a)), resulting in a sharp increase in inference latency for traditional per-token dynamic quantization methods. Although static per-tensor quantization is more efficient, it

degrades performance compared to per-token dynamic quantization. Given the substantial differences between visual and textual tokens (Fig 1 (b)), we propose modality-specific quantization (MSQ) to address the heterogeneous nature of their distributions. However, the interleaved arrangement of input tokens complicates the efficient implementation of MSQ. To address this, we further introduce the attention-invariant flexible switching (AIFS) scheme, which reorganizes tokens by placing all image tokens before text tokens. This preserves the attention equivalence while eliminating the input-dependent dynamic position overhead, thus improving computational efficiency.

**Modality-Specific Quantization.** As shown in Figure 2, our proposed modality-specific quantization (MSQ) apply different static per-tensor scaling factors to visual and textual tokens respectively. For an input sequence $E$ of length $L$ containing mixed text and visual multimodal tokens. Here, without loss of generality, we define $E = \{e_1^t, ..., e_m^v, ..., e_n^v, ..., e_L^t) \in (\boldsymbol{E}_v, \boldsymbol{E}_t)\}$, where $m$ and $n$ denote the start and end indices of the visual tokens. Then, we have:

$$E = (\overbrace{e_1^t, ..., e_{m-1}^t}^{s_t:\text{ textual scale}}, \overbrace{e_m^v, ..., e_n^v}^{s_v:\text{ visual scale}} \overbrace{e_{n+1}^t, ..., e_L^t}^{s_t:\text{ textual scale}}) \tag{7}$$

MSQ effectively addresses the substantial distributional differences between visual and textual tokens by applying distinct per-tensor scales factor for each modality. This separation not only captures the unique distribution of each modality but also prevents the significant time increase associated with token number, which grows as image and video resolutions increase. More discussions about differences between MSQ and per-tensor static and per-token dynamic quantization are in Appendix A.11, highlighting its advantages.

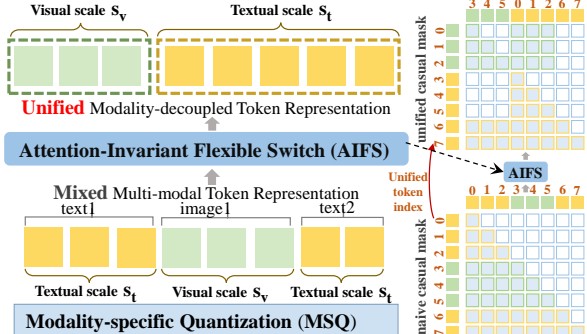

Figure 2: The depiction of modality-specific quantization (MSQ) and attention-invariant flexible switching (AIFS).

**Attention-Invariant Flexible Switching.** Due to the interleaved and non-fixed positions of visual and textual tokens in the input sequence, directly applying MSQ introduces extra and irregular data process and memory operation (e.g., slice, concat, pad), which reduces the computational efficiency of all the GEMM-based[1] layers including QK and FC. To address this issue, we propose the attention-invariant flexible switching (AIFS) scheme. AIFS rearranges the visual tokens to the front of the sequence and places the textual tokens at the end, while simultaneously adjusting the causal mask in the attention mechanism. This rearrangement preserves the computational consistency of both image and text tokens, enhancing computational efficiency. As shown in Figure 2, regardless of the input order of visual and textual tokens, the AIFS scheme consistently reorganizes the tokens in this manner.

Incorporating the AIFS scheme necessitates an adaptation of the naive causal attention. Causal attention, also known as masked self-attention (Vaswani et al., 2017), is a fundamental structure in LLMs and MLLMs due to its auto-regressive properties. It ensures that each token can only attend to or be influenced by previous tokens, preventing any influence from future tokens. The naive casual mask for token sequence $E$ can be formulated as follows:

$$\mathbf{A} = \text{Softmax}\left(\frac{\mathbf{QK}^T}{\sqrt{D}} + M_{i,j}\right), M_{i,j} = \begin{cases} 0 & \text{if } j \leq i \\ -\infty & \text{if } j > i \end{cases} \tag{8}$$

After reordering token in AIFS, it is essential to maintain the causal relationships for the unified modality-decoupled token sequence $E^u = \left\{e_m^v, ..., e_n^v, e_1^t, ..., e_{m-1}^t, ..., e_{n+1}^t, ..., e_L^t \in (\boldsymbol{E}_v, \boldsymbol{E}_t)\right\}$. Consequently, the corresponding causal mask is modified, resulting in a unified causal mask:

$$M_{i,j}^u = \begin{cases} 0 & \text{if one of the following conditions is met:} \\ & (i \leq (n-m), j \leq i \text{ or } (n-m) < j \leq n) \\ & \text{or } ((n-m) < i \leq n, (n-m) < j \leq i) \\ & \text{or } (i > n, j \leq i) \\ -\infty & \text{otherwise} \end{cases} \tag{9}$$

---

[1]GEMM (General Matrix Multiplication) is a common operation in linear algebra for matrix multiplication.

This unified causal mask preserves the inherent causal relationships among tokens, ensuring numerical equivalence during attention computations even after token reordering. Additionally, we apply corresponding changes to the position embeddings to ensure that they align with the new token indices after AIFS. This adjustment is crucial for maintaining the computation equivalence of the attention computations, as it ensures that the position information accurately reflects the revised ordering of the tokens. More details can be found in the Appendix A.

Furthermore, MSQ can be naturally integrated with AIFS to form a per-modality static quantization approach. Extensive experiments demonstrate three key advantages brought by MSQ and AIFS: **(1) Computational Equivalence and Strong Compatibility**, which can achieve SOTA quantization performance across 5 MLLMs (Table 2). **(2) Reduced Inference Latency**, achieving the speed of the per-tensor static quantization (Table 4). **(3) Enhanced Memory and Computational Efficiency**, achieving up to 24.7% speedup and 152.9% memory savings (Table 10.)

### 4.2 POST LAYERNORM-TO-RMSNORM TRANSFORMATION

Unlike the common Pre-LN structure in LLMs, MLLMs incorporate both Pre-LN and Post-LN structure in their vision encoder. As discussed in Sec 2, SliceGPT only designed a Pre-LN + Rotate Scheme for LLMs. we further develop a Post-LN + Rotate scheme to accommodate a wider range of vision encoders, enabling our MQuant to be applicable to various MLLMs in both Pre-LN and Post-LN configurations. The difference of LN style from 5 mainstream MLLMs are in Appendix 9. Here, we present how to transform post-LN into RMSNorm layer while ensuring *computational invariance*. Due to the recentering operation, LN exhibits invariance to shifts, such that $LN(\boldsymbol{X}_k - a\boldsymbol{1}) = LN(\boldsymbol{X}_k), \forall a \in \mathbb{R}$. Therefore, as shown in Figure 3. we can replace LN as RMSNorm layer through adjusting the weights $\boldsymbol{A}_2$ and bias $\boldsymbol{b}_2$ of the the linear $\ell_2$ as follows:

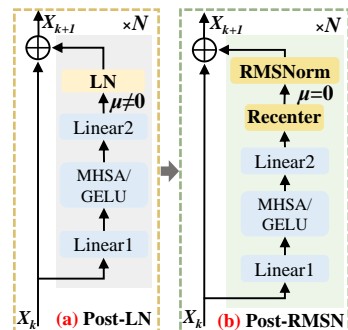

Figure 3: Post-LN to RMSNorm.

$$\hat{\boldsymbol{A}}_2 = \boldsymbol{A}_2 - \frac{1}{D}\boldsymbol{1}\boldsymbol{1}^T\boldsymbol{A}_2, \hat{\boldsymbol{b}}_2 = \boldsymbol{b}_2 - \mu(\boldsymbol{b}_2)\boldsymbol{1}, \tag{10}$$

Eq10 is the recenter operation in Figure 3 (b). Therefore, based on Eq10, the Post-LN can be replaced with an RMSNorm with the same arithmetic functionality and $\boldsymbol{X}_{k+1} = \boldsymbol{X}_k + RMSNorm(\ell_2(g(\ell_1(\boldsymbol{X}_k))))$. Ultimately, we establish the equivalence of Post-LN and Post-RMSNorm Transformers. Now that every LN in MLLMs vision encoder has been converted to RMSNorm, where we can seamlessly integrate any orthogonal matrices for handling weights and activation outliers, while ensuring that the model's output remains arithmetically equivalent.

### 4.3 ROTATION MAGNITUDE SUPPRESSION

**Theoretical Analysis of Hadamard Transformation on Weight Quantization.** (Chee et al., 2024; Tseng et al., 2024) introduced incoherence (Eq. 11) to measure quantization difficulty. Lower incoherence means easier quantization. They also showed that applying a Hadamard transform to weights and activations effectively reduces incoherence. Quarot (Ashkboos et al., 2024b) employs offline Hadamard transforms on weights and activations, along with partial online Hadamard transforms, as shown in Figure 4(a). While Quarot achieves state-of-the-art quantization results for LLMs, it does not meet expectations for MLLMs, as demonstrated in Table 2.

$$|W_{ij}| = |e_i^\top W e_j| \leq \mu \frac{\|W\|_F}{\sqrt{mn}}. \tag{11}$$

To address this, we analyze the incoherence of weight input channels before and after transformation, since per-channel quantization isn't applicable to input channels. For a weight matrix $W_{\ell_2}$, according to (Eq. 11), we can compute an incoherence coefficient $\mu_{\ell_2}$ satisfying: $\max(|W_{\ell_2}|) = \mu_{\ell_2}\frac{\|W_{\ell_2}\|_F}{\sqrt{mn}}$. Since the Hadamard transform preserves the Frobenius norm, $\|HW_{\ell_2}\|_F = \|W_{\ell_2}\|_F$. Let the incoherence coefficient of the transformed weight $HW_{\ell_2}$ be $\mu_{H\ell_2}$. Then,

$$\frac{\mu_{H\ell_2}}{\mu_{\ell_2}} = \frac{\max(|HW_{\ell_2}|)}{\max(|W_{\ell_2}|)}. \tag{12}$$

For Hadamard transform matrices, the elements in the first row and first column are all $\sqrt{\frac{1}{n}}$, and the sums of the other rows and columns are zero. For the first input channel after transformation, it always holds that $HW_{\ell_2 0j} = \sum_{r=1}^n \frac{1}{\sqrt{n}} w_{rj} = \sqrt{n} \cdot \text{mean}(w_{:,j})$. For the other input channels

after transformation, assuming the weights follow a normal distribution with mean $\mu$ and variance $\sigma^2$, the maximum value before transformation is $\mu + \sigma\sqrt{2\ln n}$, and the maximum value after transformation is $\sigma\sqrt{2\ln n}$. From the above equations, it is evident that only the first transformed input channel may cause the maximum value to increase. Therefore, when

$$\sqrt{n}\cdot\text{mean}(w_{:,j}) > \max_i w_{ij} \quad (13)$$

, the Hadamard transform will degrade the weight quantization performance. In Figure 4 (b), we show a weight from MLLMs identified using Equation 13, and its transformed distribution matches our derivations.

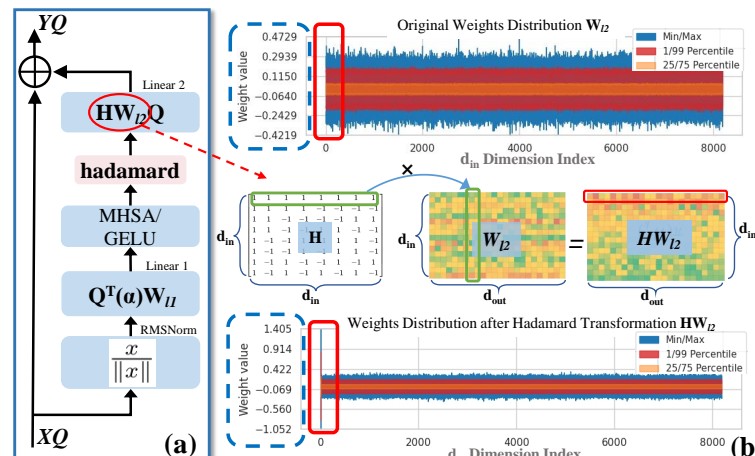

Figure 4: (a) The pipeline of Quarot, (b) transformed weights outliers.

**Rotation Magnitude Suppression for Weight Outliers.** To address the weight outliers introduced by the Hadamard transformation, we propose rotation magnitude suppression **RMS** mechanism. As shown in Figure 5, we present the quantization process for W4A8 setting. Since outliers only appear in the first row of the input channel weights, we separate this channel weights and the corresponding activations for the GEMV kernels. For the remaining weights, we perform a GEMM kernels with the original activation values. To ensure computational equivalence, we set the separated positional weights to 0. Finally, we add the results of the separated operations to obtain the output. **RMS** is a plug-and-play module with minimal overhead and, when combined with Quarot or other Hadamard-based transformations, significantly enhances quantization performance. Algorithm 2 outlines the specific procedure.

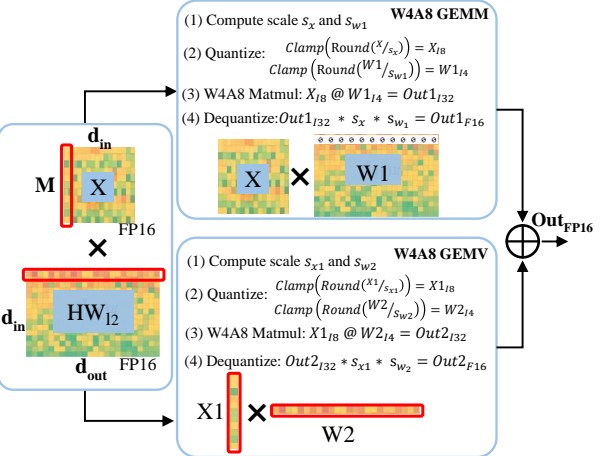

Figure 5: Schematic of RMS.

## 5 EXPERIMENTS

**Models and Datasets.** We evaluate our *MQUANT* on five MLLMs: InternVL2-8B (Chen et al., 2024a), Qwen-VL-Chat-9.6B (Bai et al., 2023b), MiniCPM-V 2.6-8B (Yao et al., 2024), Qwen2-VL-7B (Wang et al., 2024), and GLM-4V-9B (Hong et al., 2024). Evaluations are conducted on four benchmarks covering OCR and general question answering: TextVQA (Singh et al., 2019), DocVQA (Mathew et al., 2021), OCRBench (Liu et al., 2023), and MME (Fu et al., 2023), which assesses perception and cognition across 14 subtasks. These MLLMs' details are in Appendix A.9.

**Baselines and Implementation Details.** We test W8A8 and W4A8 quantization settings for both visual encoders and LLMs, comparing RTN, SmoothQuant (Xiao et al., 2022), and Quarot (Ashkboos et al., 2024b). Notably, we apply static per-tensor activation quantization for both components, unlike the dynamic per-token quantization typically used in existing MLLMs. The calibration dataset consists of 256 randomly selected samples from the corresponding benchmark training sets (Singh et al., 2019; Mathew et al., 2021; Liu et al., 2023). The batch size for latency evaluation is 1.

Table 2: Comprehensive quantization results of different MLLMs across various evaluation datasets.

| MLLMs | Method | Bits Setting Visual | LLM | TextVQA Val↑ | DocVQA Val↑ | OCRBench↑ | MME↑ |
|---|---|---|---|---|---|---|---|
| InternVL2-8B | - | BF16 | BF16 | 77.65 | 90.97 | 794 | 2209 |
| | RTN | W8A8 | W4A8 | 52.02 | 59.04 | 542 | 1528 |
| | SmoothQuant | | | 59.88 | 59.75 | 544 | 1540 |
| | Quarot | | | 73.34 | 84.07 | 715 | 2067 |
| | **MQuant (Ours)** | | | **77.49** | **90.27** | **785** | **2156** |
| | RTN | W4A8 | W4A8 | 40.06 | 31.58 | 302 | 1482 |
| | SmoothQuant | | | 46.48 | 31.21 | 310 | 1540 |
| | Quarot | | | 49.10 | 33.62 | 361 | 1941 |
| | **MQuant (Ours)** | | | **76.62** | **88.42** | **725** | **2155** |
| Qwen-VL-Chat-9.6B | - | BF16 | BF16 | 61.40 | 60.36 | 493 | 1834 |
| | RTN | W8A8 | W4A8 | 0.45 | 0.03 | 189 | 625 |
| | SmoothQuant | | | 7.45 | 7.70 | 160 | 797 |
| | Quarot | | | 45.32 | 42.44 | 286 | 940 |
| | **MQuant (Ours)** | | | **61.16** | **59.31** | **483** | **1691** |
| | RTN | W4A8 | W4A8 | 1.02 | 0.02 | 193 | 585 |
| | SmoothQuant | | | 8.59 | 4.28 | 188 | 921 |
| | Quarot | | | 46.77 | 37.35 | 289 | 1091 |
| | **MQuant (Ours)** | | | **60.50** | **58.72** | **473** | **1713** |
| MiniCPM-V 2.6-8B | - | BF16 | BF16 | 79.10 | 89.18 | 847 | 2248 |
| | RTN | W8A8 | W4A8 | 61.00 | 65.16 | 332 | 1300 |
| | SmoothQuant | | | 62.40 | 65.76 | 424 | 1510 |
| | Quarot | | | 73.71 | 80.04 | 736 | 1850 |
| | **MQuant (Ours)** | | | **80.41** | **89.15** | **844** | **2244** |
| | RTN | W4A8 | W4A8 | 60.70 | 62.23 | 351 | 1404 |
| | SmoothQuant | | | 65.67 | 60.02 | 455 | 1491 |
| | Quarot | | | 68.96 | 79.63 | 685 | 1734 |
| | **MQuant (Ours)** | | | **81.14** | **89.75** | **839** | **2189** |
| Qwen2-VL-7B | - | BF16 | BF16 | 84.43 | 93.87 | 842 | 2319 |
| | RTN | W8A8 | W4A8 | 33.92 | 52.61 | 442 | 1298 |
| | SmoothQuant | | | 49.11 | 53.97 | 444 | 1500 |
| | Quarot | | | 79.36 | 89.57 | 754 | 2045 |
| | **MQuant (Ours)** | | | **84.43** | **93.61** | **830** | **2269** |
| | RTN | W4A8 | W4A8 | 40.20 | 38.82 | 422 | 1082 |
| | SmoothQuant | | | 46.25 | 52.36 | 411 | 1535 |
| | Quarot | | | 71.44 | 83.96 | 670 | 1911 |
| | **MQuant (Ours)** | | | **84.32** | **93.58** | **824** | **2255** |
| GLM-4V-9B | - | BF16 | BF16 | 82.82 | 81.16 | 782 | 2153 |
| | RTN | W8A8 | W4A8 | 7.05 | 3.70 | 0.00 | 140 |
| | SmoothQuant | | | 9.05 | 4.10 | 0.00 | 148 |
| | Quarot | | | 82.00 | 80.17 | 782 | 2115 |
| | **MQuant (Ours)** | | | **82.06** | **80.53** | **782** | **2164** |
| | RTN | W4A8 | W4A8 | 7.61 | 3.60 | 0.00 | 163 |
| | SmoothQuant | | | 9.85 | 4.40 | 0.00 | 188 |
| | Quarot | | | 64.16 | 45.52 | 516 | 2048 |
| | **MQuant (Ours)** | | | **81.58** | **79.67** | **754** | **2120** |

## 5.1 Overall Results

**Weight-activation quantization results of various MLLMs.** MQuant can be applied to the quantization of various MLLMs. As shown in Table 2, our MQuant demonstrates significant improvements over several representative quantization methods. In W8A8 setting, MQuant achieves performance nearly equivalent to that of FP models across all evaluation datasets. Notably, even in the more challenging W4A8 setting, MQuant maintains comparable performance with FP models, while other advanced quantization methods exhibit significant performance degradation. These results indicate that our MQuant provide a more general and effective PTQ solution with strong compatibility for maintaining high accuracy in MLLMs under various quantization settings.

**Speedup and Memory Savings.** We fixed the input sequence as "text-image-text" with 15 textual tokens, varying the image resolution from $280 \times 280$ to $5600 \times 5600$. Notably, the "text-image-text" sequence setting is not arbitrarily chosen; instead, it is a common setting in existing evaluation datasets (Duan et al., 2024). We evaluate speedup and memory savings by comparing PyTorch's BF16, AWQ (W4-only), and our MQuant (W4A8). ❶ **Speedup:** As shown in Table 10, MQuant consistently achieves speedups over both PyTorch and AWQ across all resolutions, with a maximum of 24.76% over PyTorch at $840 \times 840$. Notably, MQuant outperforms AWQ, which is slower than PyTorch at most resolutions due to negative speedups. This significant speedup highlights the advantage of our

Table 3: Comparison of latency and memory saving with Pytorch (BF16), AWQ (W4-only) and ours MQuant (W4A8) on Qwen2-VL-7B. Pytorch and AWQ using the Qwen2-VL-7B official implementation. ↓ means lower values are better, ↑ means larger values are better.

| Image size | Latency (s) | | | | | Memory (G) | | | |
| H × W | Pytorch | AWQ↓ | Speedup↑ | MQuant↓ | Speedup↑ | Pytorch | AWQ↓ | Improve↑ | MQuant↓ | Improve↑ |
|---|---|---|---|---|---|---|---|---|---|---|
| $280^2$ | 0.257 | 0.286 (+0.029) | -10.14% | 0.220 (-0.037) | +16.82% | 16.45 | 7.45 (-9.00) | +120.67% | 6.50 (-9.95) | +152.92% |
| $840^2$ | 0.261 | 0.304 (+0.043) | -14.14% | 0.210 (-0.051) | +24.76% | 16.45 | 7.45 (-9.00) | +120.67% | 6.50 (-9.95) | +152.92% |
| $1400^2$ | 0.559 | 0.652 (+0.093) | -14.29% | 0.432 (-0.127) | +20.24% | 16.90 | 7.90 (-9.00) | +113.92% | 6.97 (-9.93) | +142.71% |
| $1960^2$ | 1.369 | 1.598 (+0.229) | -14.26% | 1.112 (-0.257) | +16.63% | 17.82 | 8.82 (-9.00) | +100.00% | 7.85 (-9.97) | +119.93% |
| $2520^2$ | 2.820 | 3.175 (+0.355) | -11.14% | 2.357 (-0.436) | +19.63% | 19.04 | 10.05 (-8.99) | +89.59% | 9.10 (-9.94) | +109.92% |
| $3080^2$ | 5.208 | 5.872 (+0.664) | -11.27% | 4.488 (-0.720) | +16.02% | 20.58 | 11.58 (-9.00) | +77.60% | 10.61 (-9.97) | +96.45% |
| $5600^2$ | 8.380 | 9.393 (+1.013) | -10.78% | 7.469 (-0.911) | +12.19% | 22.22 | 13.22 (-9.00) | +57.54% | 12.25 (-9.97) | +61.65% |

per-tensor static quantization, eliminating the overhead of token-wise scale computation. Even at higher resolutions (e.g., $5600^2$), MQuant maintains a 12.19% latency improvement, demonstrating scalability across various image sizes. ❷ **Memory Savings:** MQuant offers substantial memory reductions compared to both PyTorch and AWQ. It consistently reduces memory usage by over 100% compared to PyTorch (e.g., 152.92% at $840^2$) and significantly outperforms AWQ's memory efficiency, achieving up to 101.07% savings at higher resolutions. These experiments demonstrate MQuant's strengths in both latency and memory savings, achieving up to 24.76% faster inference and reducing memory consumption by over 100% compared to baselines. This significant gain is attributed to MQuant's conversion of mixed input tokens into modality-decoupled tokens, eliminating irregular memory operations (e.g., slice, concat, pad) introduced by directly applying MSQ. This transformation reduces memory consumption and mitigates the computational overhead and increased latency caused by the surge in token counts from higher image and video resolutions.

## 5.2 ABLATION STUDY

In this section, we select Qwen2-VL-7B (Wang et al., 2024), currently one of the most powerful open-source MLLMs, to ablate the effectiveness of our proposed designs.

Table 4: The accuracy and speedup of our MSQ and AIFS scheme on the linear layer during prefill stage. The input sequence is structured as "text-image-text-image-text-image-text" with an image resolution of $2240 \times 2240$ and 50 textual tokens. Latency were tested on an NVIDIA RTX 6000 Ada Generation. ↓ means lower values are better, ↑ means larger values are better.

| Activation | Weight | TextVQA Val↑ | DocVQA Val↑ | OCRBench↑ | MME↑ | Latency ↓ (s) | Speedup ↑ |
|---|---|---|---|---|---|---|---|
| BF16 | BF16 | 84.43 | 93.87 | 842 | 2319 | 1.690 | - |
| W4-g128(AWQ) | BF16 | 83.93 (-0.50) | 93.13 (-0.74) | 828 (-14) | 2252 (-67) | **2.057** (+0.367) | -17.8% |
| A8-per-token dyn | | 84.32 (-0.11) | 93.61 (-0.26) | 830 (-12) | 2269 (-50) | 1.253 (-0.437) | +34.9% |
| A8-per-tensor sta | W4-per-channel sta | 40.20 (-44.12) | 38.82 (-54.79) | 422 (-408) | 1082 (-1187) | **1.016 (-0.674)** | +66.3% |
| A8-MSQ | | 84.32 (-0.11) | 93.61 (-0.26) | 830 (-12) | 2269 (-50) | 1.085 (-0.605) | +55.8% |
| **A8-MSQ+AIFS** | | **84.32** (-0.11) | **93.61** (-0.26) | **830** (-12) | **2269** (-50) | 1.017 (-0.673) | **+66.2%** |

**Accuracy and Speedup Efficiency of the AIFS Scheme.** We compare the accuracy and speedup results of our MSQ and AIFS. As previously discussed, the number of visual tokens in MLLMs increases quadratically with image resolution, and existing dynamic per-token quantization introduces additional inference latency due to the computation of token-wise scale factors. Our AIFS transforms mixed visual and textual tokens into a unified multi-modal representation, enabling static per-tensor quantization for activations and addressing the quadratic time complexity. As shown in Table 4, MSQ+AIFS achieves the same speedup efficiency as per-tensor static quantization while maintaining near-lossless accuracy comparable to the original Float model across all linear layers in MLLMs. Furthermore,

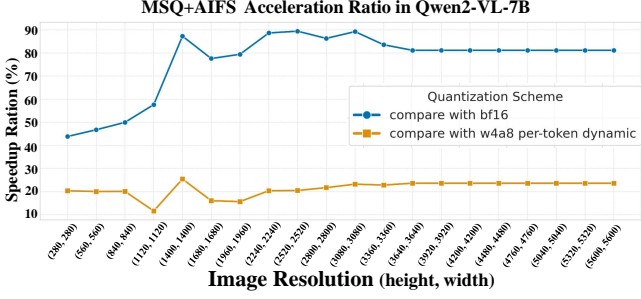

Figure 6: The Speedup of AIFS+MSQ on Qwen2-VL-7B.

we evaluate the speedip of AIFS for visual tokens under varying image resolutions, comparing it with per-token setting and FP model. As shown in Figure 6, results indicate that as image resolution increases, AIFS achieves speedups around 20% and 80%. This substantiates the effectiveness and efficiency of our MSQ and AIFS design for MLLM quantization. Moreover, since AIFS requires

only a one-time rearrangement of the input data (adjusting the causal mask and token index offline), it does not alter the overall computation graph. We also report the acceleration results both in prefill and decode stage in Table 7 of Appendix.

Table 5: Ablation study of proposed quantization designs on the OCR and MME benchmarks.

| MLLMs | Methods | | | | Bits Setting | | TextVQA Val↑ | DocVQA Val↑ | OCRBench↑ | MME↑ | Lat (ms) ↓ |
|---|---|---|---|---|---|---|---|---|---|---|---|
| | Static | AIFS + MSQ | LN2RN | RMS | Visual | LLM | | | | | |
| | | | | | BF16 | BF16 | 84.43 | 93.87 | 842 | 2319 | 6523 |
| Qwen2-VL-7B | ✓ | ✗ | ✗ | ✗ | | | 71.44 | 83.96 | 670 | 1911 | 5479 |
| | ✓ | ✓ | ✗ | ✗ | W4A8 | W4A8 | 78.95 | 87.55 | 721 | 2095 | 5484 |
| | ✓ | ✓ | ✓ | ✗ | | | 82.48 | 91.91 | 803 | 2174 | 5452 |
| | ✓ | ✓ | ✓ | ✓ | | | **84.32** | **93.58** | **824** | **2255** | 5471 |

**Ablation Study of Proposed Quantization Methods.** We conduct ablation studies on Qwen2-VL-7B (Table 5) to evaluate our quantization methods. Starting from the 'Static' baseline (both LLM and visual components quantized using GPTQ with online Hadamard transformation), we progressively incorporate our designs. Applying AIFS and MDQ to the LLM part, while keeping the visual part quantized with GPTQ, significantly improves performance, demonstrating their effectiveness for the LLM. Adding LN2RN to the visual part allows us to apply offline and online Hadamard rotation to the vision encoder, further enhancing performance. Adding RMS further achieves performance close to the FP model, showing the effectiveness of our proposed designs while maintaining high accuracy.

Table 6: Comparative quantized results under different quantization settings on the OCR and MME benchmarks. † means re-implementation based on the official weight-only quantization setting with a group size of 128 (Wang et al., 2024).

| MLLMs | Method | Bits Setting | | TextVQA Val↑ | DocVQA Val↑ | OCRBench↑ | MME↑ |
|---|---|---|---|---|---|---|---|
| | | Visual | LLM | | | | |
| | - | BF16 | BF16 | 84.43 | 93.87 | 842 | 2319 |
| | GPTQ (g128)† | BF16 | W8 | 84.33 | 93.97 | 842 | 2313 |
| | GPTQ (g128)† | BF16 | W4 | 84.18 | 93.25 | 831 | 2285 |
| Qwen2-VL-7B | AWQ (g128)† | BF16 | W4 | 83.93 | 93.13 | 828 | 2252 |
| | **MQuant** (g128) | BF16 | W4 | 84.55 | 93.18 | **832** | **2304** |
| | **MQuant** (g128) | W4 | W4 | **84.70** | **93.57** | 828 | 2292 |
| | **MQuant** (per-channel) | W4A8 | W4A8 | 84.32 | 93.58 | 824 | 2255 |

**Weight-only Quantization.** We compare our weight-only quantization results with the official Qwen2-VL-7B, which uses GPTQ (Frantar et al., 2022) and AWQ (Lin et al., 2023) to quantize only the LLM while keeping the visual encoder in BF16. For fairness, we adopt the same group size of 128. As shown in Table 6, our W4 LLM quantization aligns with their settings, achieving nearly lossless accuracy compared to other methods. Extending W4 quantization to the visual encoder enables 4-bit quantization of the entire MLLM; MQuant still performs comparably to the BF16 model and even surpasses other methods. Under the W4A8 weight-activation quantization setting, our results remain consistent, with some metrics surpassing advanced weight-only methods. These experiments confirm MQuant's effectiveness and robustness across various quantization configurations, whether for weight-only or weight-activation quantization, and for partial or full quantization of MLLMs.

## 6 CONCLUSION

In this paper, we propose *MQuant*, the first accurate and efficient post-training quantization solution for multimodal large language models (MLLMs). Addressing their unique challenges, *MQuant* reduces the time to first token (TTFT) with per-tensor static quantization and introduces modality-specific quantization (MSQ) to handle distribution discrepancies between visual adn textual tokens. Besides, we propose attention-invariant flexible switching (AIFS) to ensure consistent positioning of multimodal tokens and computational invariance, enhancing quantization efficiency. Furthermore, we propose a unified LayerNorm to RMSNorm transformation, enabling seamless integration of the MLLM vision encoder with Hadamard rotation. Finally, we reveal the weight outliers caused by the online Hadamard rotations and propose Rotation Magnitude Suppression (RMS) to mitigate them. Extensive experiments on five mainstream MLLMs demonstrate that *MQuant* attains state-of-the-art PTQ performance, with the W4A8 model matching FP model accuracy. These findings highlight *MQuant*'s potential to advance MLLM quantization and enhance practical applicability.

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

## A APPENDIX

### A.1 ROTARY POSITION EMBEDDING FOR ATTENTION-INVARIANT FLEXIBLE SWITCHING

Many modern LLMs (Touvron et al., 2023a;b; Dubey et al., 2024) use rotary position embedding (RoPE) (Su et al., 2021) to encode information about the order of tokens in the input sequence. Rotary position embeddings are linear transformations applied to keys and queries defined as:

$$\mathbb{R}_{\Theta,m}^{d_h} = \begin{pmatrix} \cos i\theta_1 & -\sin i\theta_1 & 0 & 0 & \cdots & 0 & 0 \\ \sin i\theta_1 & \cos i\theta_1 & 0 & 0 & \cdots & 0 & 0 \\ 0 & 0 & \cos i\theta_2 & -\sin i\theta_2 & \cdots & 0 & 0 \\ 0 & 0 & \sin i\theta_2 & \cos i\theta_2 & \cdots & 0 & 0 \\ \vdots & \vdots & \vdots & \vdots & \ddots & \vdots & \vdots \\ 0 & 0 & 0 & 0 & \cdots & \cos i\theta_{d_h/2} & -\sin i\theta_{d_h/2} \\ 0 & 0 & 0 & 0 & \cdots & \sin i\theta_{d_h/2} & \cos i\theta_{d_h/2} \end{pmatrix} \quad (14)$$

where $i \in [1, L]$ is the token index, $\Theta = \{\theta_i = 10000^{-2(i-1)/D}, i \in [1, 2, ..., D/2]\}$, and $\theta_i, i \in 1..D/2$ are predefined constants.

In the proposed Attention-Invariant Flexible Switching (AIFS) mechanism, we also apply the rearrangement for position embedding to maintain the computation equivalence. For a mixed input token $E = \{e_1^t, ..., e_m^v, ..., e_n^v, ..., e_L^t) \in (\boldsymbol{E}_v, \boldsymbol{E}_t)\}$ (in Eq 7), where $m$ and $n$ denote the start and end indices of the visual tokens. Specifically, after AIFS, the unified token can formulated as: $E = \{e_m^v, ..., e_n^v, e_1^t, ..., e_{m-1}^t, ..., e_L^t) \in (\boldsymbol{E}_v, \boldsymbol{E}_t)\}$. Therefore, the unified token indices after AIFS can be represented as:

$$(m, ..., n, 1, ..., m-1, n+1, ..., L) = AIFS(1, ..., m, ..., n, ..., L) \quad (15)$$

Due to we are aware of the final indices for input token after AIFS, than we can utilize the reorder token indices for Eq 14 to get the corresponding position embedding.

### A.2 SPEEDUP OF MSQ+ AIFS.

| Stage | BF16 | Per-token Dynamic | Ours | Ours + GEMV | Speedup |
|---|---|---|---|---|---|
| Prefill | 1690 | 1253 | 1017 | - | **+23%** |
| Decode | 17.5 | 16.4 | 13.06 | 8.2 | **+100%** |

Table 7: Latency (ms) comparison with Float and per-token dynamic quantization in W4A8 setting.

Here, we provide a more comprehensive comparison, we further report the acceleration results on the decode stage. Configuration is aligned with Table 4, measuring mean latency (ms) of linear layer for decoding 2,000 tokens. A custom kernel was implemented for W4A8 kernel GEMV operations. As shown in Table 7, compared to per-token dynamic quantization, in addition to achieving **23%** speed improvement during the prefill stage, our method achieves an **100%** speed up in decode stage. Overall, our AIFS+MSQ transforms the time-consuming online dynamic quantization into offline static quantization, achieving significant acceleration with almost no loss in accuracy, especially in long sequences. Notably, in practical applications, using OpenAI's token pricing as an example, our method can save ≈30% in costs, and this effect is even more pronounced in other MLLMs, as visual tokens are more expensive than textual tokens.

### A.3 WEIGHTS OUTLIERS AFTER ONLINE HADAMARD TRANSFORMATION

Following (Tseng et al., 2024; Ashkboos et al., 2024b) we make use of fast Hadamard transformation where convenient. Hadamard matrix is an orthogonal matrix with entries proportional to $\{+1, -1\}$.

A Walsh-Hadamard matrix is a square matrix of size $2^n$ with For a Hadamard matrix:

$$\boldsymbol{H}_2 = \frac{1}{\sqrt{2}} \begin{bmatrix} 1 & 1 \\ 1 & -1 \end{bmatrix} \qquad \text{and} \qquad \boldsymbol{H}_{2^n} = \boldsymbol{H}_2 \otimes \boldsymbol{H}_{2^{n-1}}. \tag{16}$$

$$(AB)_{ij} = \sum_{r=1}^{n} a_{ir} b_{rj} = a_{i1} b_{1j} + a_{i2} b_{2j} + \cdots + a_{in} b_{nj} \tag{17}$$

Thereby the $\boldsymbol{H}W_{\ell_2}$ can be formulated as:

$$(\boldsymbol{H}W_{\ell_2})_{ij} = \sum_{r=1}^{n} h_{ir} w_{rj} = h_{i1} w_{1j} + h_{i2} w_{2j} + \cdots + h_{in} w_{nj} \tag{18}$$

where $\boldsymbol{H} \in \mathbb{R}^{d_{in} \times d_{in}}$ and $W_{\ell_2} \in \mathbb{R}^{d_{in} \times d_{out}}$, $d_{in}$ and $d_{out}$ is the dimension of input and output of weight $W_{\ell_2}$. Due to the properties of the Hadamard matrix $\boldsymbol{H}$, whose first row consists entirely of 1, for the first row in $(\boldsymbol{H}W_{\ell_2})$, $(\boldsymbol{H}W_{\ell_2})_{0j} = \sum_{r=1}^{n} w_{rj}$, due to the property of Hadamard matrix $\boldsymbol{H}$. So, the values in $W_{\ell_2}$ are subject to continuous accumulation and summation, resulting in the exists of outliers in the first row of the output matrix $\boldsymbol{H}W_{\ell_2}$. Notably, the matrix $\boldsymbol{H}W_{\ell_2}\boldsymbol{Q}$ still has the same problem, for simplicity, we omit the matrix $\boldsymbol{Q}$ in the main paper.

### A.4 IMAGE TOKENS IN VARIOUS MLLMS

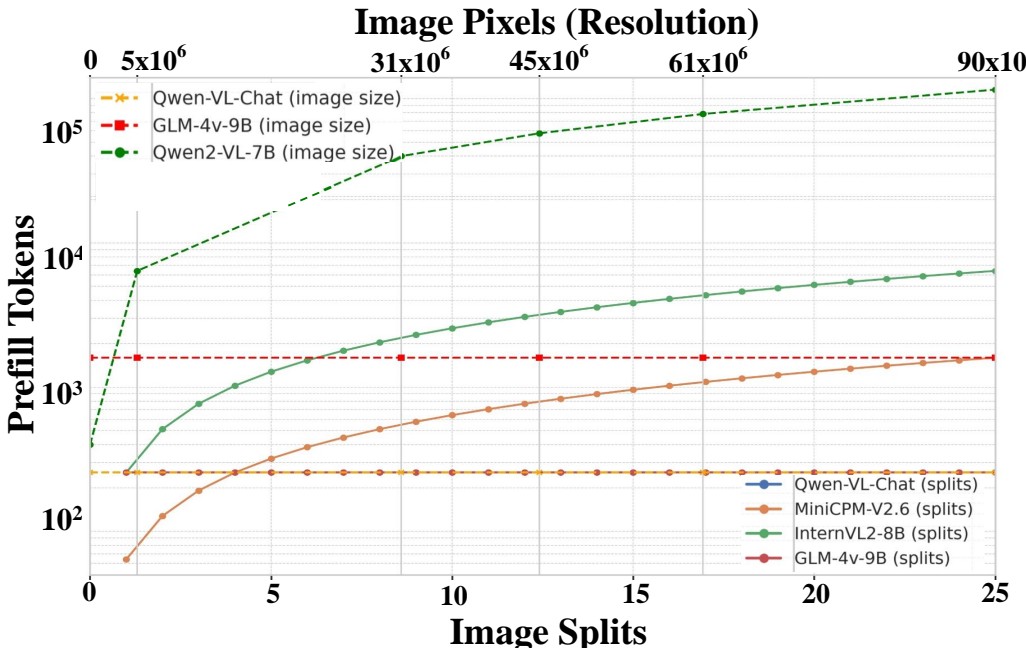

Figure 7: The number of prefill visual tokens across different MLLMs as the image splits or resolution increases.

As shown in Fig 7, for different MLLMs (Bai et al., 2023b; Wang et al., 2024; Yao et al., 2024; Chen et al., 2024b), the number of prefill visual tokens grows as image resolution increases. This rapid expansion in token count exacerbates the inference latency, particularly in per-token dynamic quantization, which requires separate processing for each token, such as memory access and scale computation. As a result, the TTFT in MLLMs increases drastically, severely impacting overall inference latency. Moreover, in higher-demand scenarios such as video-based tasks and multi-image dialogues, the accumulation of visual tokens becomes even more pronounced, further exacerbating the increase in TTFT.

### A.5 MQUANT ALGORITHM

Here, we present our *MQuant* algorithm for MLLMs in Algorithm 1.

---

**Algorithm 1** *MQuant* Quantization Algorithm

---

**Input**: Full-precision (FP) MLLM model with a vision encoder $E$, visual-language projector $P$, and a large language model **LLM**; Calibration dataset $D^c$.

**Output**:

- For $E$ and $P$: per-channel weight scale $s_w^E$, per-channel weight zero-point $z_w^E$, per-tensor activation scales $s_a^E$, per-tensor activation zero-point $z_a^E$.

- For **LLM**: per-channel weight scale $s_w^{llm}$, per-channel weight zero-point $z_w^{llm}$, per-tensor activation scales $s_{a^v}^{llm}$ for visual tokens and $s_{a^t}^{llm}$ for textual tokens, per-tensor activation zero-points $z_{a^v}^{llm}$ for visual tokens and $z_{a^s}^{llm}$ for textual tokens.

1: Apply Hadamard Rotation to the LLM Part as described:
2: Apply the offline and online Hadamard rotations to all the weights and activations in **LLM**.
3: Quantize Weights of the LLM:
4: Input the calibration dataset $D^c$ to the FP MLLM.
5: Use GPTQ to quantize the weights for **LLM**, obtaining per-channel weight quantization parameters $s_w^{llm}$ and $z_w^{llm}$.
6: For the LLM Part:
    (a) Input $D^c$ to the FP MLLM.
    (b) Compute per-tensor activation quantization parameters $s_{a^v}^{llm}$ and $s_{a^t}^{llm}$, $z_{a^v}^{llm}$ and $z_{a^t}^{llm}$ for visual and textual tokens respectively, based on the proposed Modality-Specific Quantization (MSQ) in Sec 4.1.
    (c) Reorder the input mixed token sequence from $E = e_1^t, \ldots, e_m^v, \ldots, e_n^v, \ldots, e_L^t$ to a unified modality-decoupled sequence $E^u = e_m^v, \ldots, e_n^v, e_1^t, \ldots, e_{m-1}^t, \ldots, e_{n+1}^t, \ldots, e_L^t$ using the proposed Attention-Invariant Flexible Switching (AIFS) scheme in Sec **??**.
7: Transform all the LayerNorm to RMSNorm in MLLM vision encoder $E$ and Visual-Language Projector $P$ using the proposed Post LayerNorm-to-RMSNorm transformation in Sec 4.2.
8: Apply the offline and online Hadamard rotations to all the weights and activations in $E$ and $P$.
9: Quantize Weights of $E$ and $P$:
10: Input $D^c$ to the transformed FP vision encoder $E$ and Visual-Language Projector $P$.
11: Use GPTQ to quantize $E$ and $P$, obtaining per-channel weight quantization parameters $s_w^E$ and $z_w^E$.
12: Address the weight outliers using caused by online Hadamard based on the proposed Rotation Magnitude Suppression (RMS) in Sec 4.3.

---

## A.6 Effectiveness of Rotational Magnitude Suppression for Weight Outliers in LLMs.

For LLMs, compared to the original Quarot, integrating RMS with Quarot leads to performance improvements across LLaMA2 models with 7B, 13B, and 70B parameters, as detailed in Table 8. For MLLMs, ablation studies presented in Table 5 demonstrate that the RMS method significantly enhances quantization performance.

Table 8: WikiText-2 PPL on 4-bit quantization of LLAMA-2 models with 2048 sequence length.

| Method | Weight Quantization | LLAMA-2 7B | LLAMA-2 13B | LLAMA-2 70B |
|--------|--------------------|-----------|-------------|-------------|
| Baseline | - | 5.47 | 4.88 | 3.32 |
| QuaRot | GPTQ | 6.10 | 5.40 | 3.79 |
| QuaRot | RMS + GPTQ | **6.04** | **5.32** | **3.67** |

## A.7 Weight Outliers from on Rotation

Previous study Quarot (Ashkboos et al., 2024b) leveraged incoherence processing (Chee et al., 2023; Tseng et al., 2024) to make the weights matrix and activations easier to quantize by multiplying randomized Hadamard matrix. Here, we briefly summarize the pipeline of Quarot for LLMs. As shown in Figure 8 (a), we select a randomized Hadamard ma-

trix with size that matches the hidden dimension of the model, denoted as $Q$. **(1)** the input $X$ is incoherence process to enable improved quantization, becoming $X \leftarrow XQ$; **(2)** we remove the scaling operation from RMSNorm $(\text{diag}(\alpha))$ and absorbing into the subsequent weight matrices $W_{\ell_1}$ of input linear; **(3)** to cancel the impact of $Q$, $Q^\top$ is applied to weight matrices $W_{\ell_1}$, then $W_{\ell_1} \leftarrow Q^\top(\alpha)W_{\ell_1}$; **(4)** to improve the quantize-ability of the activations within each block, insert online Hadamard operations before the output linear $\ell_2$; **(5)** to retain the computational invariance, Hadamard matrix $H$ is implicitly reversed by fusing into the output linear matrix $W_{\ell_2}$:

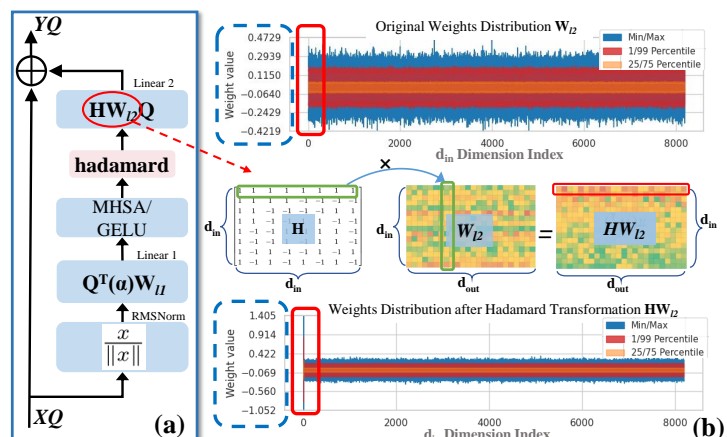

Figure 8: (a) The pipeline of Quarot, (b) transformed weights outliers.

$W_{\ell_2} \leftarrow HW_{\ell_2}Q$. Without loss of generality, the weight matrices $W_{\ell_1}$ can also be query/key/value-projection matrix, and the weight matrices $W_{\ell_2}$ can also be output weight matrix, please refer to Quarot for more quantization details.

## A.8 RMS ALGORITHM

Here, we present our *RMS* design with Quarot in Algorithm 2.

---

**Algorithm 2** RMS Integration with Quarot

---

**Require:** An LLM or MLLM model
**Ensure:** Quantized model with RMS
 1: Initialize an empty list $marks$
 2: **for** each linear layer in the $model$ **do**
 3:     **if** Layer satisfies Equation 13 **then**
 4:         Mark the layer and append its ID to $marks$
 5:     **end if**
 6: **end for**
 7: Apply Quarot or other Hadamard-based transformations to the $model$
 8: **for** each layer ID in $marks$ **do**
 9:     Modify the layer's implementation using the RMS method
10: **end for**
11: Quantize the $model$
12: **return** $model$

---

## A.9 COMPARISON OF DIFFERENT MLLMs: INPUT PRE-PROCESS, LAYERNORM ARCHITECTURE IN VISION ENCODER, MODEL PARAMETERS AND FLOPS.

In this section, we compare the visual input pre-process methods, LayerNorm structures in MLLM vision encoder, model parameters, and Flops across five mainstream MLLMs: InternVL2-8B (Chen et al., 2024a), Qwen-VL-Chat-9.6B (Bai et al., 2023b), MiniCPM-V 2.6-8B (Yao et al., 2024), Qwen2-VL-7B (Wang et al., 2024), and GLM-4V-9B (Hong et al., 2024). This comparison highlights the architectural differences between these models, particularly the TTFT sensitivity to input image resolution, illustrating the unique challenges these variations present for quantization and efficient inference.

## A.10 SPEEDUP AND MEMORY SAVINGS WITH SCALING IMAGE RESOLUTION

We fixed the input sequence as "text-image-text" with 15 textual tokens and presente the detailed changes of speed and memory, varying the image resolution from $280 \times 280$ to $5600 \times 5600$. Notably,

Table 9: Comparison of TTFT sensitivity to image resolution, model Parameters and flops in mainstream MLLMs. $^\dagger$ means the Flops values are measured with the number of visual tokens is 256.

| MLLMs | TTFT's sensitivity to input image resolution | LayerNorm | Params (B) | | FLOPs (T)$^\dagger$ | |
|---|---|---|---|---|---|---|
| | | | Visual | LLM | Visual | LLM |
| InternVL2-8B | TTFT increases with input image aspect ratio | Pre-LN | 0.34 | 7.74 | 1.28 | 7.54 |
| Qwen-VL-Chat-9.6B | Fixed input resolution (448×448) | Pre-LN | 1.94 | 7.73 | 4.23 | 3.70 |
| MiniCPM-V 2.6-8B | TTFT increases with input image aspect ratio | Pre-LN | 0.49 | 7.61 | 4.16 | 3.64 |
| Qwen2-VL-7B | TTFT increases **quadratically** with input image resolution | Pre-LN | 0.68 | 7.61 | 1.31 | 3.61 |
| GLM-4V-9B | Fixed input resolution (1120×1120) | Post-LN | 4.51 | 8.78 | 12.10 | 4.70 |

Table 10: Comparison of latency and memory saving with Pytorch and AWQ on Qwen2-VL-7B. $^\ddagger$ means the Qwen2-VL-7B official implementation.

| Image size | Pytorch$^\ddagger$ (BF16) | | AWQ$^\ddagger$ (W4-only) | | MQuant (W4A8) | |
|---|---|---|---|---|---|---|
| H × W | Latency(s) | Memory(G) | Latency(s) | Memory(G) | Latency(s) | Memory(G) |
| $280^2$ | 0.257 | 16.45 | 0.286 (-10.14%) | 7.45 (+120.67%) | 0.220 (**+16.82%**) | 6.50 (**+152.92%**) |
| $560^2$ | 0.252 | 16.45 | 0.292 (-13.70%) | 7.45 (+120.67%) | 0.211 (**+19.48%**) | 6.50 (**+152.92%**) |
| $840^2$ | 0.261 | 16.45 | 0.304 (-14.14%) | 7.45 (+120.67%) | 0.210 (**+24.76%**) | 6.50 (**+152.92%**) |
| $1120^2$ | 0.326 | 16.58 | 0.384 (-15.10%) | 7.56 (+119.51%) | 0.262 (**+24.48%**) | 6.61 (**+151.59%**) |
| $1400^2$ | 0.559 | 16.90 | 0.652 (-14.29%) | 7.90 (+113.92%) | 0.432 (**+20.24%**) | 6.97 (**+142.71%**) |
| $1680^2$ | 0.881 | 17.33 | 1.066 (-17.39%) | 8.33 (+108.57%) | 0.705 (**+18.23%**) | 7.40 (**+130.27%**) |
| $1960^2$ | 1.369 | 17.82 | 1.598 (-14.26%) | 8.82 (+100.00%) | 1.112 (**+16.63%**) | 7.85 (**+119.93%**) |
| $2240^2$ | 2.013 | 18.40 | 2.294 (-12.24%) | 9.40 (+95.74%) | 1.653 (**+17.83%**) | 8.44 (**+117.84%**) |
| $2520^2$ | 2.820 | 19.04 | 3.175 (-11.14%) | 10.05 (+89.59%) | 2.357 (**+19.63%**) | 9.10 (**+109.92%**) |
| $2880^2$ | 3.883 | 19.77 | 4.345 (-10.64%) | 10.77 (+83.81%) | 3.297 (**+17.69%**) | 9.82 (**+101.07%**) |
| $3080^2$ | 5.208 | 20.58 | 5.872 (-11.27%) | 11.58 (+77.60%) | 4.488 (**+16.02%**) | 10.61 (**+96.45%**) |
| $3360^2$ | 6.814 | 21.46 | 7.548 (-9.73%) | 12.46 (+70.65%) | 6.004 (**+13.41%**) | 11.50 (**+83.01%**) |
| $3640^2$ | 8.360 | 22.22 | 9.377 (-10.84%) | 13.22 (+57.54%) | 7.469 (**+11.91%**) | 12.25 (**+61.65%**) |
| $4480^2$ | 8.349 | 22.22 | 9.379 (-10.97%) | 13.22 (+57.54%) | 7.469 (**+11.71%**) | 12.25 (**+61.65%**) |
| $5600^2$ | 8.380 | 22.22 | 9.393 (-10.78%) | 13.22 (+57.54%) | 7.469 (**+12.19%**) | 12.25 (**+61.65%**) |

the "text-image-text" sequence setting is not arbitrarily chosen; instead, it is a common setting in existing evaluation datasets (Duan et al., 2024). We evaluate speedup and memory savings by comparing PyTorch's BF16, AWQ (W4-only), and our MQuant (W4A8). **Speedup:** As shown in Table 10, MQuant consistently achieves speedups over both PyTorch and AWQ across all resolutions, with a maximum of 24.76% over PyTorch at $840 \times 840$. Notably, MQuant outperforms AWQ, which is slower than PyTorch at most resolutions due to negative speedups. This significant speedup highlights the advantage of our per-tensor static quantization, eliminating the overhead of token-wise scale computation. Even at higher resolutions (e.g., $5600^2$), MQuant maintains a 12.19% latency improvement, demonstrating scalability across various image sizes. **Memory Savings:** MQuant offers substantial memory reductions compared to both PyTorch and AWQ. It consistently reduces memory usage by over 100% compared to PyTorch (e.g., 152.92% at $840^2$) and significantly outperforms AWQ's memory efficiency, achieving up to 101.07% savings at higher resolutions. These experiments demonstrate MQuant's strengths in both latency and memory savings, achieving up to 24.76% faster inference and reducing memory consumption by over 100% compared to baseline methods. This makes MQuant a more efficient solution for deploying MLLMs in resource-constrained environments.

## A.11 QUANTIZATION GRANULARITY

Furthermore, as mentioned in SoomthQuant (Xiao et al., 2022), there are different different granularity levels. The **per-tensor static** quantization uses a single step size for the entire matrix. **Per-token dynamic** quantization employs different $s$ for the activations associated with each token, being a common granularity for activations quantization of existing LLMs. For weights, per-channel quantization applies distinct $s$ for each output channel of the weights, while group-wise quantization utilizes a coarse-grained $s$ for different channel groups. Notably, group-wise quantization is a prevalent granularity for weight quantization in LLMs (Frantar et al., 2022; Yao et al., 2022). Please refer to appendix for more quantization basics.

## A.12 ADVANTAGE OF MSQ AND AIFS

In per-tensor static quantization, the quantization parameters (i.e., scale and zero-point) are precomputed for an entire tensor (e.g., weights or activations) and remain fixed throughout inference. While efficient, this approach often leads to large and unacceptable accuracy loss in MLLMs due to their diverse activation distributions across varying inputs.

In contrast, per-token dynamic quantization computes quantization parameters on-the-fly for each input token during inference. This approach incurs significantly higher computational overhead, as the quantization parameters must be recalculated for every input token, along with multiple additional memory traversals. Such requirements make per-token dynamic quantization unfriendly or impractical for edge devices and some AI accelerators, which struggle with fine-grained dynamic operations Tan et al. (2024). This issue is especially severe in MLLMs, where the token count increases significantly with higher image resolution or more video frames. The Modality-Specific Quantization (MSQ) in MQuant is a novel per-modality quantization approach specifically designed to address the unique challenges of MLLMs quantization.

Furthermore, MSQ can be naturally applied to the unified modality-decoupled tokens generated by AIFS. By integrating MSQ and AIFS, our designs yields three key advantages: **(1) Computational Equivalence and Strong Compatibility**: The unified causal mask and token index introduced by AIFS preserves the inherent causal relationships among tokens, ensuring numerical equivalence during attention computations. Moreover, since AIFS requires only a one-time rearrangement of the input data (adjust causal mask and token index in offline), it does not alter the overall computation graph. This characteristic allows for seamless integration with other LLM inference acceleration methods, such as FlashAttention (Dao et al., 2022), ensuring both computational equivalence and strong compatibility. As shown in Table 2, MQuant achieves SOTA quantization performance across 5 mainstream MLLMs. **(2) Reduced Inference Latency**: MSQ not only addresses the substantial distributional differences between modalities but also mitigates the significant computational overhead and increased inference latency caused by the surge in token counts from higher image and video resolutions. As shown in Table 4, MSQ+AIFS significantly reduces latency from 2.057s to 1.1017s, closely matching the speed of the per-tensor static setting while maintaining near-lossless accuracy comparable to the original Float model. **(3) Enhanced Memory and Computational Efficiency**: By combining MSQ and AIFS, we convert mixed input tokens into unified, modality-decoupled tokens, eliminating the irregular memory operations (e.g., slice, concat, pad) introduced by directly applying MSQ. This transformation reduces memory consumption and improves efficiency of GEMM kernel, which would otherwise be compromised by the interleaved and non-fixed positions of visual and textual tokens. As shown in Table 10, MQuant can achieve up to 24.7% speedup and 152.9% memory savings.

## A.13 LAYERNORM, RMSNORM AND COMPUTATIONAL INVARIANCE

We introduce LayerNorm, RMSNorm, the computational Invariance, and their usage in Transformers.

**Layer Normalization** (LayerNorm, LN) (Ba, 2016) is a technique to normalize the activations of intermediate layers of neural networks. Given a vector $x \in \mathbb{R}^d$, LayerNorm normalizes it to obtain a zero-mean unit-variance vector,

$$\text{LayerNorm}(x) = \frac{x - \mu(x)\mathbf{1}}{\sqrt{x_2^2/d - \mu^2(x) + \epsilon}}, \text{where } \mu(x) = \frac{\mathbf{1}^T x}{d}, \epsilon > 0. \tag{19}$$

LayerNorm recenters and rescales the activations and gradients in the forward and backward computations, which enables fast and robust training of neural networks.

**Root Mean Square Normalization** (RMSNorm) (Zhang et al., 2019) is another technique used for normalizing the activations. It is similar to LayerNorm in that it aims to accelerate and stabilize the training but uses a different normalization approach. Instead of normalizing the inputs based on their mean and variance, RMSNorm normalizes them based on their root mean square (RMS) value. It is defined in the following equation,

$$\text{RMSNorm}(x) = \frac{x}{\sqrt{x_2^2/d + \epsilon}}, \text{where } \epsilon > 0. \tag{20}$$

RMSNorm only rescales the input vector and the corresponding gradients, discarding the recentering process. As shown in their definitions, RMSNorm is computationally simpler and more efficient

than LayerNorm. It is reported that replacing LayerNorm with RMSNorm can achieve comparable performance and save training and inference time by $7\% - 64\%$ (Zhang et al., 2019).

Given a zero-mean vector $x$, these two kinds of normalization are equivalent. Formally, if $\mu(x) = 0$, then LayerNorm$(x)$ = RMSNorm$(x)$. We may optionally introduce learnable parameters and apply an element-wise affine transformation on the output of LayerNorm and RMSNorm.

**LayerNorm (LN) and RMSNorm**    Given the input concated token $X$ after embeddings with the shape $L \times D$, the $X$ is passed through a LayerNorm (Ba, 2016) operation, which subtracts the mean from each row of the matrix, divides the row by its standard deviation, rescales (columnwise), and adds an offset. Follow (Ashkboos et al., 2024a), we write the LayerNorm block as

$$\text{LayerNorm}(X) = \text{RMSNorm}(XM)\text{diag}(\alpha)\sqrt{D} + \mathbf{1}_N \beta^\top \qquad (21)$$

where RMSNorm$(X)$ applies $x \leftarrow x/\|x\|$ to each row of $X$, and $X = concat(E_v, E_t))$ is the concatenation between text tokens $E_t$ and the visual tokens $E_v$. The vector parameter $\alpha$ and offset (vector) parameter $\beta$ are learned independently at each LayerNorm instance. The constant matrix $M = I - \frac{1}{D}\mathbf{1}\mathbf{1}^\top$ is a $D \times D$ matrix which subtracts the mean from each row of $X$, called recentering operation. Formally, if $M = I$, the input $X$ has a zero-mean, the Eq 21 is equivalent to RMSNorm. Specifically, LayerNorm is widely employed in visual encoders $E$, whereas RMSNorm (Zhang et al., 2019) is commonly used in LLMs (Touvron et al., 2023a; Dubey et al., 2024) and has been shown to accelerate training and inference time with similar performance (Zhang et al., 2019).

**Computational Invariance in RMSNorm.**    Based on the *computational invariance*, recent studies (Ashkboos et al., 2024a;b) have shown that orthogonal transformations can effectively smooth outliers and improve the quantize-ability of both weights and activations. In particular, for transformers, inserting linear layers with an orthogonal matrices $Q$ before and after the RMSNorm (Zhang et al., 2019) layer in a transformer, the network remains unchanged. In detail, given the input $X$ and orthogonal matrix $Q$ for RMSNorm layer, the *computational invariance* means: RMSNorm$(XQ)Q^\top$ = RMSNorm. Here, $Q^\top Q = QQ^\top = I$ and a rotation matrix is an orthogonal matrix with $|Q| = 1$. Note that multiplying a vector $x$ by $Q$ does not change the norm of the vector, since $\|Qx\| = \sqrt{x^\top Q^\top Q x} = \sqrt{x^\top x} = \|x\|$.

A.14 LAYERNORM TO RMSNORM TRANSFORMATION.

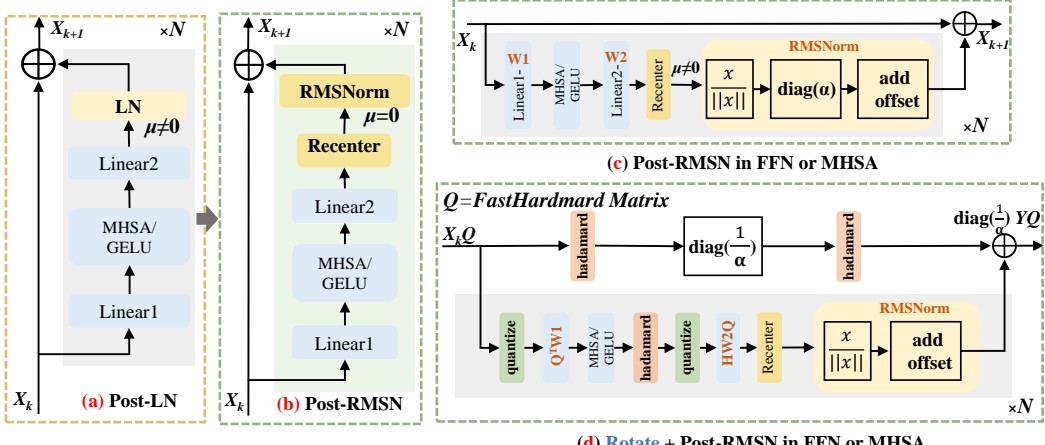

Figure 9: The proposed Post-LN + Rotate Scheme.

**Post-LayerNorm to RMSNorm Transformation.**    As shown in Figure 9, we present the detailed Post-LN + Rotate design.

**Pre-LayerNorm to RMSNorm Transformation.**    Here, we propose unified LayerNorm-to-RMSNorm transformation, aiming to synthesize the transformer architecture of MLLMs' vision

encoders and LLMs, endowing them with rotation-friendly characteristics that facilitate the effective removal of outliers. We take Pre-LN transformer as an example to show that how transform Pre-LN into RMSNorm layer while guaranting arithmetic equivalence. As shown in Figure 10 (a), for the input $X_k$ of the $k$-th block, the main block in pre-LN transformer is $X_{k+1} = X_k + \ell_2(g(\ell_1(LN(X_k))))$, where $k \in [1, N]$, and $N$ is the block number. If $g$ is an activation function, such as GELU, this block is a multi-layer perceptron (MLP) module. If $g$ is a multi-head attention, then this block is the casual attention module (Vaswani et al., 2017). Due to the recentering operation, LN exhibits invariance to shifts, such that $LN(X_k - a\mathbf{1}) = LN(X_k), \forall a \in \mathbb{R}$. Therefore, as shown in Figure 10 (b), we can replace LN as RMSNorm layer through two modifications:

❶ recenter the input $X_k$ to $X_k - \mu(X_k)\mathbf{1}$, ensuring that the input to norm layer maintain a zero mean. ❷ adjust the weights $A_2$ and bias $b_2$ of the the linear $\ell_2$ to $\hat{A}_2 = A_2 - \frac{1}{D}\mathbf{1}\mathbf{1}^T A_2, \hat{b}_2 = b_2 - \mu(b_2)\mathbf{1}$. Consequently, the LN can be replaced with an RMSNorm layer with the same arithmetic functionality. The first operation is to recenter $X_k$, while the second operation is to recenter the output of main branches. Notably, since $X_{k+1} = X_k + \ell_2(g(\ell_1(LN(X_k))))$, after applying ❶ and ❷, the input and the output of main branch are re-centered with zero-mean, while the input of residual branches also maintain a zero mean. Therefore, the output after current blocks, $X_{k+1}$ (which serves as the input for next block), still maintain zero-mean. A detailed proof is provided in the Appendix. Ultimately, we establish the equivalence of Pre-LN and Pre-RMSNorm Transformers. Now that every LayerNorm in the transformer has been converted to RMSNorm in MLLMs, we can use any orthogonal matrices $Q$

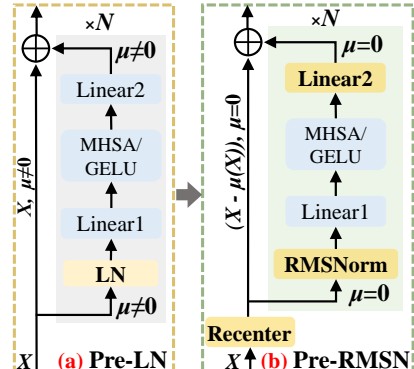

Figure 10: The illustration of transformation from Pre-LN to RMSNorm.

to the model. Therefore, the visual encoder and LLMs are in a rotation-friendly RMSNorm-only transformer architecture.

### A.15 TRANSFORMER FORWARD PASS

Here, we refer from SliceGPT (Ashkboos et al., 2024a) to describe the transformer forward pass process. As described in Figure 10 illustrates part of a transformer network: an attention block connected to a Feed Forward Network (FFN) block through a LayerNorm or RMSNorm block, with residual connections.

**Embeddings** Let $D$ be the embedding dimension of our transformer, $N$ be the sequence length. The transformer model takes as input a sequence of token IDs and position IDs, and uses them to index the embedding matrices, producing the initial signal $X$ with shape $N \times D$. In what follows we consider, without loss of generality, a single embedding matrix $W_{embd}$ indexed by input sequence $s$.

**Attention Blocks in LLMs** The attention block has four matrices: $W_k, W_q, W_v$ and $W_o$, each of dimension $D \times D$. The input signal arriving into the block is projected into the Key ($XW_k$), Query ($XW_q$), and Value ($XW_v$) matrices, which are then split into multiple heads. A nonlinear operation is applied at each head before the signals are combined and multiplied by the output weight matrix $W_o$. Since the first three weight matrices are applied separately to the inputs, we can concatenate them and perform a single matrix multiplication. We can consider the concatenation of these matrices to be a single linear layer, which we denote as Linear1 $\ell_1$, as shown in Figure 10. The MHSA in Figure 10 means the attention computation. We also refer to the output linear as Linear2 $\ell_2$, as shown in Figure 10. We treat the attention block as MHSA($XW_1$)$W_2$, where MHSA represents the multi-head attention operation.

**FFN Blocks** The other type of block that appears in transformer architectures is a Feed Forward Network (FFN) block. In many cases, this is a Multi-layer Perceptron (MLP), which also can be represented with Figure 10. A Linear1 layer $\ell_1$ with weight matrix $W_1$, followed by an non-linear operation (i.e., GELU function), and Linear2 layer: $\ell_2$ with weight matrix $W_2$. We can therefore denote the operation of MLP or gated FFN layers as GELU($XW_1$)$W_2$.

**Language Modelling (LM) Head** All of the transformer networks to which we apply in this paper have a decoder-only structure: after multiple layers applying alternating attention and FFN blocks, a

head block computes logits which are used to compute the loss during training and token prediction on deployment. The head operation is $\boldsymbol{X}\boldsymbol{W}_{\text{head}} + \boldsymbol{b}_{\text{head}}$.

Once the model is trained and all of the parameters are set, the computations required in a transformer network to produce predictions involve passing signal matrices from one block to the next until the head node is reached. Since we are able to define both FFN and attention blocks in the form $\sigma(\boldsymbol{X}\boldsymbol{W}_1)\boldsymbol{W}_2$, where we understand that $\sigma$ represents either a point-wise (GELU function) or multi-head-attention nonlinearity (MHSA), we are able to describe the forward pass using Algorithm 3.

---

**Algorithm 3** Pre-LN Transformer

---

**Require:** $\{\boldsymbol{W}_1^\ell, \boldsymbol{W}_2^\ell\}_{\ell=1}^L$      weights and biases of FFN and attention blocks
**Require:** $\{\sigma_\ell\}_{\ell=1}^L$      nonlinearity associated with each block
**Require:** $\{\text{Norm}_\ell\}_{\ell=0}^L$      LayerNorm or RMSNorm instances to perform between blocks
**Require:** $\boldsymbol{W}_{\text{embd}}, \boldsymbol{W}_{\text{head}}, \boldsymbol{b}_{\text{head}}$      embedding and head matrices
**Require:** $\boldsymbol{s}$      input sequence

 1: $\boldsymbol{X} \leftarrow \boldsymbol{W}_{\text{embd}}[\boldsymbol{s}, :]$      index embeddings
 2: $\boldsymbol{X} \leftarrow \text{Norm}_0(\boldsymbol{X})$      normalize
 3: **for** $\ell = 1 \ldots L$ **do**
 4:     $\boldsymbol{Z} \leftarrow \sigma_\ell(\boldsymbol{X}\boldsymbol{W}_{\text{in}}^\ell)\boldsymbol{W}_2^\ell$      apply FFN or attention
 5:     $\boldsymbol{X} \leftarrow \text{Norm}_\ell(\boldsymbol{X} + \boldsymbol{Z})$      normalize and apply residual connection
 6: **end for**
 7: **return** $\boldsymbol{X}\boldsymbol{W}_{\text{head}} + \boldsymbol{b}_{\text{head}}$      apply model head

---

