# OpenReview forum: "MQuant: Unleashing the Inference Potential of Multimodal Large Language Models via Full Static Quantization"
_ICLR.cc/2025/Conference — Submitted to ICLR 2025_

### Official Review · Reviewer_Gy4o · 2024-10-30

**Soundness:** 3
**Presentation:** 3
**Contribution:** 3
**Rating:** 6
**Confidence:** 2

**Summary:**

This paper proposes MQuant, an accurate and efficient post-training quantization solution for multimodal large language models (MLLMs). MQuant reduces the time to first token (TTFT) with per-tensor static quantization and introduces modalityspecific quantization (MSQ) to handle distribution discrepancies between visual and textual tokens. Experiments on five mainstream MLLMs demonstrate that MQuant attains state-of-the-art PTQ performance.

**Strengths:**

Strength：

1. Extensive experiments demonstrate the approach's effectiveness in the PTQ of MLLMs.
2. The motivation is clear and quantization for MLLM is an important topic.
3. This paper is well-organized and clearly-written.

**Weaknesses:**

Weakness:

1. My only concern is that i'm not familiar with quantization. So i will adjust my rating depending on the other reviewers' opinions.

**Questions:**

Please see the comments above.

---

> ### Author Response · Authors · 2024-11-21
>
> Sincerely thank you for your feedback!
>
> If you have any further questions or would like clarification on any specific points related to our work, we would be more than happy to engage in further discussions. Your insights are valuable to us, and we are here to help.
>
> If there are still any doubts, please feel free to let us know, and we will make every effort to solve them.

---

### Official Review · Reviewer_FtkM · 2024-11-04

**Soundness:** 3
**Presentation:** 3
**Contribution:** 2
**Rating:** 6
**Confidence:** 3

**Summary:**

This paper proposes a quantization method which is specifically tailored towards MLLM. Because of the distributional differences between visual tokens and text tokens, the authors intuitively calculate separate quantization scales for two modalities and calibrate the attention mask accordingly. Further, they adapt some techniques from the LLM quantization literature to visual encoders in MLLM. By combining these two, MQuant maintains lower performance degradation under challenging quantization settings on multiple state-of-the-art retrained MLLM models.

**Strengths:**

1. The paper follows an intuitive approach to study MLLM quantization. The authors identify the issues based on some observations in the experiments and resolve the problem in a step-by-step manner.
2. The efficacy of the method is supported by extensive experiments.  The paper shows the quantization performance of 5 mainstream MLLM models on various multi-modal tasks. The ablation studies demonstrate the usefulness of different components in maintaining the performance near the float-point baseline.

**Weaknesses:**

1. The delivery of the paper needs significant improvement. The text is highly redundant.
- Introduction: The content of the second last paragraph mostly overlap the main contribution part. It could be beneficial if these two parts are reorganized or condensed.
- Methodology: In 4.1, there are abundant words to explain the reason why we need MSQ and AIFS and the benefits brought by these two. To me, these are intuitive and simple operations which only need concise words for explanation. For 4.2 and 4.3, which are the techniques adapted from LLM quantization, it would be better if the authors could emphasize their novel improvements or adaptations rather than putting too many words to explain other people's contributions.
- Although using separate figures for different components are informative, it could be easier for the readers to follow without reading the algorithm 1 in Appendix first if the authors could add a figure to show the overall quantization pipeline with the novel parts highlighted.
- For some abbreviations used in the paper, like GEDD and W4A8, it would be friendly to readers not in the area if adding the explanations in the first place.

2. The paper does not demonstrate enough novelty.  First, both LayerNorm-to-RMSNorm transformation and Hadamard rotation are borrowed from LLM quantization literature (Ashkboos et al., 2024a, b). Second, although adopting a simple Divide-and-Conquer strategy like paper does to cope with the distribution outliers or differences may be sufficient, it is worth thinking about other systematic alternatives after getting more insights from the observations in the experiments. For now, the paper is more like a technical report. The paper should be concise and highlight the actual novel contributions.

3. Experiments: It would be better to see the latency comparisons among the proposed quantization methods could be added in Table 5.

4. Minor Errors:

- The font size of the legend in Figure 1 (left side) is too small to read.
- Line 85-87: the meaning of the sentence Is not clear. Two "slightly" exist.
- For Table 3/4. the arrow directions showing the relative difference are counter-intuitive. Showing the decrease of latency with down arrows and adding "lower is better" could be an alternative.
- In Table 5, should that be "MSQ" rather than "MDQ"?

**Questions:**

1. In Eq (6), should the denominator of equation $s$ be $2^b-1$? since for b-bit, the value range would be (0,  $2^b-1$).
2. In line 321, "easier to quantize". What does easy mean in this context?
3. In line 287, what do the "outliers" mean? Extremely low or high values?

---

> ### Author Response · Authors · 2024-11-21
>
> Thank you for reviewing our work and providing useful suggestions. Please check our detailed reply to your questions/comments.
>
> **Q1**: Paper Writing.
>
> **A1**: 1. **Introduction**: We have refined the content of the introduction and contributions based on your suggestions, further highlighting the core contributions of our method. In the introduction, we aim to present low-level description and rigorous logic to explain how our proposed method systematically addresses the issues encountered in MLLM quantization. In the contributions, we provide a more general summary of our key contributions, including the unique observation analysis of MLLM quantization that led to the development of MSQ, AIFS, Post-LN+Rotate scheme and RMS module, as well as the effectiveness achieved by our method.
>
> 2.**Method**: Respectfully disagree that the method is redundant.
> * In Section 4.1, the motivation for our MSQ and AIFS methods is aimed at reducing the Time to First Token (TTFT) while avoiding additional computational burden. Therefore, we believe it is necessary to clearly articulate the advantages brought by our methods and provide a thorough analysis.
>
> * In Section 4.2, we propose an equivalent transformation for the **Post-LN transformer structure** that differs from the existing method SliceGPT. SliceGPT only discusses how to convert the Pre-LN transformer structure to RMSNorm. **Our unified LN-to-RMSNorm enabes our Mquant to be effective and general for both Pre-LN- and Post-LN-based MLLM approaches**. Particularly, we presented the **different LN styles of various MLLM models in Table 7 in Appendix**. This distinction is a key contribution, as our method demonstrates generalizability to various LayerNorm structures.
>
> * In Section 4.3, our contribution focuses on analyzing the root causes of weight outliers and proposing effective solutions. Thus, describing online Hadamard rotation and analyzing the emergence of weight outliers is essential. Based on our in-depth analysis, we present a simple and effective solution, Rotation Magnitude Suppression (RMS), which addresses a unique problem not yet presented in existing works and constitutes one of our core contributions.
>
> **Q2**: abbreviation.
>
> **A2**: The GEDD means the GEMM？**GEMM** (General Matrix Multiplication) is a widely used operation in linear algebra that performs matrix multiplication. **W4A8** (Weight 4-bit Activations 8-bit) refers to a quantization scheme used in neural networks, where weights are represented with 4 bits and activations with 8 bits. We have clearly defined these terms upon their first occurrence in the manuscript and have updated them in the revised manuscript.
>
> **Q3**: Defend our novelty
>
> **A3**:
> * Our research is rooted in a deep exploration on the unique quantization issues in MLLMs and provides a comprehensive analysis based on these valuable observations, revealing the root causes of performance collapse during MLLM quantization (speed limitation of dynamic per-token, data distribution differences of multi-modal input, sensitive outliers).
>  * To facilitate efficient inference for variable-sequence input tokens, we propose **Modality-specific Quantization (MSQ) and Attention-Invariant Flexible Switching (AIFS)** to support per-tensor static quantization while maintaining lossless accuracy.
>  * To ensure the generalization of our Mquant across various MLLMs, we propose an equivalent transformation from **Post-LN + Rotate scheme**, distinguishing from SliceGPT which only presents pre-LN + Rotate scheme.
>  * We further identified weight outlier magnitudes caused by Hadamard rotation and proposed **Rotation Magnitude Suppression (RMS)** to mitigate it.
>   * Extensive results across five different MLLMs demonstrate the effectiveness and generalizability of our Mquant, which is, to the best of our knowledge, **the first efficient and accurate PTQ solution for MLLMs**.
>   * More importantly, as discussed above, our approach can achieve **objective economic cost savings** in practical deployments and provides valuable insights for the application of MLLMs on edge devices.
>
> **Q4** Table 5 in Experiments.
>
> **A4** Thanks for your suggestions. We have added the latency in Table 5 and updated it in the new version of the manuscript.
>
> **Q5** Minor Errors
>
> **A5**
> 1. Thanks for your carefully reading. To make Figure 1 more clear, we added a new big figure in the appendix to provide a clearer presentation.
> 2. Line 85-87 have been rewritten and colored as blue.
> 3. In Table 3 and 4, we aim to demonstrate the **relative performance improvements** of our Mquant and existing methods AWQ in terms of latency and memory compared to floating-point models. Upward arrows (↑) indicate positive improvements, while downward arrows (↓) indicate negative improvements. To present more clearly, we changed the arrows to '+' and '-' symbols.

---

> > ### Comment · Reviewer_FtkM · 2024-11-24
> >
> > Thank the authors for replying to my comments.
> >
> > 1. With regard to method section, I did not mean that the structure is redundant. The section is well organized with enough information in it but the content or text should be concise. For section 4.1, I am not convinced that it is useful to put lots of words in stating the advantages of the proposed techniques (line 273-287) without verification of experimental results. It would be better to demonstrate the efficacy of your method in experiment section. Besides, for each of subsection 4.1-4.3, I believe that there is room for text to be condensed. I acknowledge that it is necessary to introduce the existing methods for analyzing the issues, but please make sure that details about previous methods, like layernorm to RMSNorm or Hadamard Matrix, be reduced while your own novel contributions should be highlighted.
> >
> > 2. For Table 3 and 4, I understand that you want to highlight the performance improvement. But the current form is slightly counterintuitive. I would recommend showing the vanilla arithmetic differences, i.e., the memory changing from 22.22 to 13.22 ($\textbf{-}$57.54%) and adding the explanation "lower is better" in the caption.

---

> ### Author Response · Authors · 2024-11-21
>
> **Q6** Equation 6
>
> **A6** Thanks for you feedback! In our experiment, we utilize signed uniform quantization, so the scale is defined as follows:
> $$
> s = \frac{\max(|\mathbf{x}|)}{2^{b-1} - 1}
> $$.We have updated Equation.6 accordingly.
>
> **Q7** easier to quantize
>
> **A7** Thank you for your question. In this context, "**easier to quantize**" refer to the weight and activation distributions being more **uniform and there are no significant outliers**. These characteristics allow for the application of existing, naive post-training quantization (PTQ) methods with minimal adjustments. As a result, the quantization process does not introduce excessive quantization error, thereby avoiding a significant drop in performance.
>
> **Q8**：outliers
>
> **A8**： In the context of LLM/MLLM quantization, **outliers** refer to values in the weight or activation distributions that are significantly different from the majority of the data points, typically represented by extremely high values. This is a common issue for LLM quantization[1,2,3,4,5]. Outliers can have a large impact on the quantization process, leading to issues such as:
>   * Quantization Error: Outliers can skew the calculation of scaling factors and zero points, making it difficult to represent the range of values accurately. This can result in a higher quantization error.
>   * Performance Degradation: If outliers are not handled properly, they can lead to significant drops in model performance after quantization, as the quantized model may struggle to handle inputs that were originally represented well by the floating-point model.
>
> If there are still any unresolved doubts, please feel free to let us know, and we will make every effort to solve them.
>
> [1.] SmoothQuant: Accurate and Efficient Post-Training Quantization for Large Language Models, ICML 2023
>
> [2.] GPTQ: ACCURATE POST-TRAINING QUANTIZATION FOR GENERATIVE PRE-TRAINED TRANSFORMERS ICLR2023.
>
> [3.] Outlier Suppression+: Accurate quantization of large language models by equivalent and optimal shifting and scaling, EMNLP 2023.
>
> [4.] PB-LLM: Partially Binarized Large Language Models, ICLR 2024
>
> [5.] QuaRot: Outlier-Free 4-Bit Inference in Rotated LLMs, arxiv 2024.

---

> ### Author Response · Authors · 2024-11-25
>
> Dear reviewer  FtkM :
>
> We want to thanks for your positive discussion. Please check our detailed reply below.
>
> **Q1**: The clarification of our writing in Sec 4.1 (MSQ+AIFS).
>
> **A1**:  1. Regarding line 273-287 in section 4.1, we **respectfully disagree with the assertion that we have not validated the advantages of our proposed MSQ+AIFS**. In contrast, we have provided comprehensive experiments that demonstrate the effectiveness of MSQ+AIFS. (Please see Table 2, 3, 4, 5 in Experiments).
>
> 2. **Clarifying the Goal of MSQ and AIFS**: Our proposed MSQ and AIFS aim to achieve the same accuracy as per-token dynamic quantization while reaching the speed of per-tensor static quantization.  In MLLM quantization, **per-tensor static quantization achieves the fastest inference speed (speed upper bound)**, but it leads to **significant performance loss**. Although **per-token dynamic quantization performs well (accuracy upper bound)**, the online token-wise computation of scales **limits the MLLM's inference speed.** (Please refer to ***General Comments 1*** for background information regarding per-token dynamic quantization and per-tensor static quantization.)
>
> | Method | Linear Latency (s) ↓ |Speedup↑ | TextVQA Val | DocVQA Val | OCRBench | MME |
> |---|---|---|---|---|---|---|
> | per-token dynamic | 1.253 (baseline) | - | 84.32 | 93.61 | 830 | 2269 |
> | per-tensor static | 1.016 | **+23%** | 40.20 (**-44.12**) | 38.82 (**-54.79**) | 422 (**-408**) | 1082 (**-1187**) |
> | MSQ | 1.085 |**+16%** | 84.32 | 93.61 | 830 | 2269 |
> | AIFS+MSQ | 1.017 |**+23%** | 84.32 | 93.61 | 830 | 2269 |
>
> 3. Our proposed MSQ and AIFS aim to achieve the same accuracy as per-token dynamic quantization while reaching the speed of per-tensor static quantization. We have updated Table 4, presenting both speed and accuracy results, and plotted a figure in this anonymous link https://ibb.co/ZB4kKSq. Our MSQ + AIFS achieves speeds nearly on par with per-tensor static quantization while attaining the accuracy of per-token dynamic quantization.
>
> 4. Here is the detailed experiments which have included in the original manuscript to demonstrate the advantages in line 273-287 one-by-one.
> * Reduced Inference Latency: As demonstrated in Table 4, MSQ+AIFS significantly **reduces latency from 2.057s to 1.1017s** , closely matching the speed of the per-tensor static setting.
> * Computational Equivalence and Strong Compatibility: We utilize the Eq. 7, 8,9 in sec 4.1, along with Eq 11, 12 in Appendix A.1 to **demonstrate the computation equivalence** of AIFS. And the comprehensive and general experiments across five mainstream MLLMs presented in Table 2 further **illustrate the strong compatibility** of our MSQ+AIFS.
> * Enhanced Memory and Computational Efficiency: Table3 demonstates significant improvements in speed and memory savings, achieving up to **24.7% speedup and 152.9% memory savings**.
>
> **Q2**: The clarification of our writing in Sec 4.2 and 4.3.
>
> **A2**: 1. **Regarding sec 4.2**, we think it is necessary to introduce preliminaries (Line290-296) such as computational equivalence before detailing our proposed Post-LN + Rotate scheme. This introduction helps clarify the motivation and intricacies of our method for the reader. This is also similar to the writing style of published method[1], which dedicates a a section to computational equivalence as preliminaries.
>
> 2. **Regarding Sec 4.3**, our contribution extends beyond merely proposing Rotation Magnitude Suppression; we conduct an in-depth analysis of the root causes of anomalous weight outliers and provide corresponding solutions. Therefore, it is essential to explain how online Hadamard rotation impacts activations and weights, leading to weight outliers. After thoroughly analyzing this issue, we propose Rotation Magnitude Suppression (RMS) as a targeted solution.
>
> **Q3**: The Table 3 and 4.
>
> **A3**:  Thanks for your suggestion. We revise and present **the vanilla arithmetic differences** in Table 3 and 4 to help understand the advantages of speed and performance of our MQuant.  All changes in manuscript are marked with blue.
>
> If there are still any unresolved doubts, please feel free to let us know, and we will make every effort to solve them.
>
> [1] QuaRot: Outlier-Free 4-Bit Inference in Rotated LLMs, NeurIPS 2024

---

> > ### Comment · Reviewer_FtkM · 2024-11-27
> >
> > Thank the authors for the follow-up reply. I was not "asserting" that the advantages of MSQ+AIFS module have not been verified. I would suggest that the analyzes about the advantages in line 273-287 be moved to experiment section after reporting the relevant results just like how you articulated in A1. In this way, it seems to me that these statements could become more solid rather than be separated from experiments. Again, for the whole method section, I would argue that the writing could be further improved by condensing the text about previous methods and highlighting your own contributions.
> >
> > Overall, this paper is a good technical paper with clear motivations derived from the issues found in the experiments. Thank the authors for the efforts in providing additional experiment results and considering my comments. I would like to raise my score to 5.

---

> ### Author Response · Authors · 2024-11-29
> **Thanks for your recognition**
>
> Dear Reviewer FtkM,
>
> Thank you for your constructive comments and valuable suggestions. We greatly appreciate the time and effort you dedicated to reviewing our manuscript. Your feedback on the writing significantly contributed to improving the quality of our paper. We extend our sincere gratitude!
>
> **Q1**：Advantages in lines 273-287
>
> **A1**：According to your suggestion, we've revised the manuscript to move the analysis from lines 273-287 in **Section 4.1** to the Experiment section (specifically, just after reporting the relevant results in **Sections 5.1 and 5.2**).
> - This reorganization ensures that the advantages of MSQ and AIFS are immediately supported by experiments, making the conclusions more solid and allowing readers to better appreciate the practical impact of our contributions. **Thanks!**
>
> **Q2**：Method writing
>
> **A2**：We appreciate your insights regarding the balance between discussing prior work and emphasizing our contributions.
> - **Changes Made**: We have thoroughly revised the method in **Sections 4.2 and 4.3** to condense the discussion of previous methods except for the necessary context, focusing more on our core contribution.
> - **Highlighting Our Contributions**: We have restructured the section to more prominently feature our novel contributions:
>   - We streamline the description of Post-LN+Rotate in **Section 4.2** and add a detailed schematic of the Post-LN+Rotate scheme in **Appendix A.14** to highlight our core contribution.
>   - We add a detailed theoretical analysis identifying the root cause of why online Hadamard rotations can lead to quantization degradation due to significant weight outliers. **This** analysis not only identifies the problem but also justifies the need for our proposed **Rotation Magnitude Suppression (RMS)** method in **Section 4.3**. Besides, we also added algorithms and generalization experiments of RMS for LLMs (Table 8) in **Appendices A.7 and A.8** to show its effectiveness.
>
> Thank you again for your thoughtful review. We hope that these revisions adequately address your concerns and **enhance your confidence in the contributions of our work**. We believe that, **after incorporating your suggestions as well as those from the other reviewers, the latest version represents a substantial improvement over the one you evaluated last time** (28 Nov 2024). This also allows readers to more readily understand the significance of our contribution and how it advances the field. We are committed to advancing the field of MLLM quantization and providing valuable insights to the community.
>
> Therefore, we kindly ask if you could take some time to review our improvements and provide a re-evaluation. (All the changes are colored blue in the revised manuscript.)
>
> Sincerely,
>
> The Authors

---

> > ### Comment · Reviewer_FtkM · 2024-11-30
> >
> > Thank the authors for considering my suggestions and refine paper writing accordingly. I will raise my score to 6.

---

> ### Author Response · Authors · 2024-12-02
>
> Dear Reviewer FtkM
>
>
> Thank you very much for your recognition and support of our work! We greatly appreciate the time and effort you have dedicated to reviewing our manuscript. Your suggestions have played a significant role in enhancing the overall quality and clarity of the final paper. We will keep moving to make MQuant and the following projects better and better.
>
>
> Sincerely,
>
> The Authors

---

### Official Review · Reviewer_xqur · 2024-11-04

**Soundness:** 3
**Presentation:** 3
**Contribution:** 2
**Rating:** 5
**Confidence:** 3

**Summary:**

This paper introduces several techniques to enhance the accuracy and reduce the inference latency of Multimodal Large Language Models (MLLMs), which are affected by the additional vision encoder/adaptor. Empirical results demonstrate that the quantized model obtained using the proposed method outperforms other quantization methods in terms of accuracy and inference speed under certain settings.

**Strengths:**

1. The paper is well-written and easy to follow.
2. The modality-specific quantization and Layernorm-to-RMSNorm transformation are well-motivated by the distributional differences of various modality modules and architectural designs.
3. Comprehensive experimental results are provided on various MLLMs, with comparisons to several popular recent LLM quantization methods.

**Weaknesses:**

1. Attention-Invariant Flexible Switching (AIFS) Scheme: The authors claim that the proposed AIFS scheme is computationally equivalent to the original attention computation. However, it is unclear whether the corresponding positional embeddings are adjusted accordingly. If not, the equivalence may not be ensured.

2. Experiment Settings: There are concerns regarding the experimental settings. In Section 5.1, the authors conducted experiments under the "text-image-text" setting with 15 textual tokens. However, inference settings can be more complex:
- In a batch, the number of textual tokens varies, resulting in different attention masks after AIFS.
- There can be interleaved image-text inference with more image-text turns.
- There can also be multi-image inference with single or multiple turns.
More clarifications under these cases are required to further show the efficacy of the proposed method.

**Questions:**

1. For the proposed AIFS scheme, are the positional embeddings adjusted accordingly as the attention mask changes?
2. What batch sizes were used when evaluating the inference latency?

---

> ### Author Response · Authors · 2024-11-21
> **Position embedding also adjusts in AIFS and add more test cases**
>
> Thank you for reviewing our work and providing useful suggestions. Please check our detailed reply to your questions/comments.
>
> **Q1**: position embeddings in AIFS.
>
> **A1**: In AIFS, we also **apply corresponding changes to the positional embeddings** to ensure that they align with the new token indices after AIFS. Since we are aware of the changes in token indices before and after AIFS, we can apply the adjustment for position embedding to maintain the computation equivalence. This adjustment is crucial for maintaining the numerical equivalence of the attention computations, as it ensures that the positional information accurately reflects the revised ordering of the tokens. **More details are in Appendix A**.
>
>
> **Q2**: More complex  inference settings:
>
> **A2**: 1.Our "text-image-text" sequence setting is not arbitrarily chosen; rather, it is a common setting in existing evaluation datasets [1]. Therefore, we selected it for evaluation.
>
> * As you mentioned, there are indeed more complex dialogue scenarios in practical applications, such as multi-turn conversations or multi-image reasoning. It is the root motivation that we propose MSQ and AIFS (Sec 1.), aiming to address the online token-wise scale computation and memory storage issues associated with dynamic per-token quantization.
>
> 2. Here, we also report a more comprehensive multi-modal input tokens configuration, utilizing a "text-image-text-image-text-image-text" sequence setting. In this configuration, each text is represented as a **token sequence of length 300**, while the number of tokens corresponding to each **image continuously increases from 100 to 10000**. The number of tokens in the **decode phase** is uniformly fixed at a length of **2,000 tokens**. As the number of tokens per image increases (i.e., with higher image resolution), this effectively corresponds to multi-image inference. Besides, the mainstream Qwen2-VL-7B-Instruct supports a **maximum input token number of 32,768**, and our configuration (8) approaches this upper limit.
>
> | |text-Image-text-Image-text-image-text|Prefill Stage| | |Decode Stage (Generate 2K tokens)| | |
> |-|-|-|-|-|-|-|-|
> | | | FP16 | ours |Speedup| FP16 |ours |Speedup|
> |(1)|300-100-300-100-300-100-300|0.90|0.67|**+34.3%**|51.02|33.4|**+52.5%**|
> |(2)|300-400-300-400-300-400-300|0.96|0.69|**+39.1%**|53.82|36.12|**+49.0%**|
> |(3)|300-1600-300-1600-300-1600-300|1.24|1.01|**+22.8%**|58.04|40.55|**+43.1%**|
> |(4)|300-2500-300-2500-300-2500-300|2.12|1.72|**+23.3%**|64.92|46.71|**+39.0%**|
> |(5)|300-3600-300-3600-300-3600-300|3.35|2.81|**+19.2%**|66.52|48.93|**+35.9%**|
> |(6)|300-4900-300-4900-300-4900-300|4.98|4.22|**+18.0%**|68.21|50.36|**+35.4%**|
> |(7)|300-6400-300-6400-300-6400-300|7.09|5.95|**+19.2%**|75.31|58.16|**+29.5%**|
> |(8)|300-10000-300-10000-300-10000-300|13.57|11.23|**+20.8%**|101.55|83.13|**+22.2%**|
>
> * The above token setup can encompass both interleaved image-text inference with single or multi-turns. Experiments demonstrate that our method achieves up to **39.1% and 52.5%** speed improvements in the prefill and decode phases.
>
> * Although the acceleration effect diminishes as the input token count increases, this aligns with the observations in Sage-Attention [2], where the **primary computational overhead arises from attention operations** as the number of input tokens grows, thereby weakening the acceleration advantage of the linear layers. However, even when approaching the maximum multi-modal input token limit, our method still provides **~20%** speedup. Notably, to ensure fairness, we did not quantize the KV cache during the decoding phase. If the KV cache were to be quantized, it could yield better acceleration results, further demonstrating that our method is **orthogonal to existing KV cache quantization** methods [3, 4].
>
> **Q3**: attention masks after AIFS in a batch
>
> **A3**:  Yes, in a batch, each token sequence requires a corresponding causal mask. Specifically, since AIFS requires only a one-time rearrangement of the input data (by adjusting the causal mask and token index offline), it does not alter the overall computation graph. This characteristic allows for seamless integration with other LLM inference acceleration methods, ensuring both computational equivalence and strong compatibility.
>
> **Q4**: batch sizes for inference latency?
>
> **A4**: In the evaluation of inference latency, we set the batch size to 1. Experiment details have been updated in the paper.
>
> If there are still any unresolved doubts, please feel free to let us know, and we will make every effort to solve them.
>
> [1] Vlmevalkit: An open-source toolkit for evaluating large multi-modality models, ACM MM 2024.
>
> [2] SageAttention: Accurate 8-Bit Attention for Plug-and-play Inference Acceleration, Arxiv 2024.
>
> [3] KIVI: A Tuning-Free Asymmetric 2bit Quantization for KV Cache, ICML 2024.
>
> [4] KVQuant: Towards 10 Million Context Length LLM Inference with KV Cache Quantization，NeurIPS 2024.

---

> ### Author Response · Authors · 2024-12-02
> **Add Multi-batch and Multi-turn Experiments**
>
> Dear  Reviewer xqur
>
> We hope this message finds you well. We would like to sincerely thank you for your thoughtful review and valuable feedback on our paper. Your constructive comments have been instrumental in helping us improve our work.
>
> Here, we further present the speedup for **multi-batch and multi-turns dialogue inference** of our MQuant. We utilize the Qwen2VL-7B-Instruct model with an NVIDIA RTX 6000 Ada.
>
> **1. Multi-batch Inference**
> * We report a more multimodal input tokens configuration **when batch size > 1**, utilizing a "text-image-text" sequence setting with an image resolution of 2240 × 2240 and **variable textual tokens (from 50 to 200 in different batch channels)**. The number of tokens in the **decode phase** is uniformly fixed at a length of 512 tokens. We use this configuration to report the inference acceleration results when batch size > 1. During multi-batch inference, we first identify the longest token length within the batch. Subsequently, we left-pad the shorter sequences with `pad_token_id` to align all batches to this maximum length. By applying left padding, the padding tokens are associated with the image modality. Additionally, the padded regions are assigned a mask value of 0, ensuring that they do not interfere with attention computations and thereby do not affect the final results. For clarity, we also plot an illustration of causal mask when batch size >1 in this anonymous link https://ibb.co/F4rG27x.
>
> |**Batch**||**config (Text+Image+Text)**||**Prefill(s)**| |**↑Speedup** |**Decode(s)**| | **↑Speedup**|**All Network(s)**|  |**↑Speedup**|
> |:-:|:-:|:-:|:-:|:-:|:-:|:-:|:-:|:-:|:-:|:-:|:-:|:-:|
> | |**Text**|**Image**|**Text**|**bfp16**|**MQuant(ours)**| |**bfp16**|**MQuant(ours)** |  |**bfp16**|**MQuant(ours)**|  |
> |1|10|2240x2240|50|2.54|1.93|31.6%|18.01|12.89|39.7%|20.55|14.82|38.7%|
> |2|10/10|2240x2240/2240x2240|50/100|5.42|4.15|**+30.6%**|37.82|31.56|**+19.8%**|43.24|35.71| **+21.1%**|
> |3|10/10/10|2240x2240/2240x2240/2240x2240|50/100/150|8.24|6.42|28.3%|48.03|40.35| 19.0%|56.27|46.77|20.3%|
> |4|10/10/10/10 |2240x2240/2240x2240/2240x2240/2240x2240|50/100/150/200|11.17|8.67 |28.9%|59.09|49.92|18.4%|70.26|58.59|20.0%|
>
> (1) **The whole network speed**:
>   * As shown in the table above, we present the acceleration effects of **multi-batch inference** with batch sizes ranging from 1 to 4. Compared to the FP model, experiments demonstrate that our MQuant achieves speed improvements of **~20%** during the whole prefill and decode stages **when batch size > 1**.
>
> (2) **Linear-only speed**:
> |stage|fp16|Dynamic W4A8|Ours|Ours+GEMV|Speedup than FP|Speedup than Dynamic|
> |:-:|:-:|:-:|:-:|:-:|:-:|:-:|
> |Prefill|1690|1253|1017|-|**+66%**|**+23%**|
> |Decode|17.5|16.4|13.06|8.2|**+113%**|**+100%**|
>
> * Here, we also report the speed up of our MQuant on the **linear layer** during the **prefill and decode stage**. Here, configuration is aligned with Table 4, measuring mean latency (**ms**) of linear layer for decoding 2,000 tokens. A custom kernel was implemented for W4A8 kernel GEMV operations.
>
>  As shown in the table, compared to the FP model，our MQuant achieves **66%** and **113%** speed improvements during the prefill and decoding stages, respectively. Even when compared to per-token dynamic quantization, MQuant achieves **23%** and **100%** speed improvement.
>
> **2. Multi-turns Inference**
>
> We present a multi-turns dialogue configuration, utilizing a "text-image-text" sequence setting with an image resolution of 2240 × 2240 and textual tokens is 50 in each dialogue turn.  During the decoding phase, the number of tokens in each dialogue turn is uniformly fixed at 512 tokens. Additionally, we store the key-value caches and position IDs for each turn to facilitate the multi-turn dialogue experiments.
>
> |Turns| |config in a turn| |All(s)| |Speedup|
> |:-:|:-:|:-:|:-:|:-:|:-:|:-:|
> | |Text|Image|Text|bfp16|ours| |
> |1|10|2240x2240|50|20.55|14.82|**+38.7%**|
> |2|10|2240x2240|50|44.06|32.61|**+35.1%**|
> |3|10|2240x2240|50|76.67|59.48|**+28.9%**|
>
> As shown in the table above, we present the acceleration effects of **multi-turn inference** with rounds ranging from 1 to 3. Compared to the FP model, experiments demonstrate that our MQuant achieves up to **38.7%** in speed improvements in the whole prefill and decode stages. The above experiments in multi-batch and multi-turns all demonstrate the efficiency and generality of our MQuant. Notably, to ensure fairness, we did not quantize the KV cache in the above experiments. All of the above experiments will be included in the final paper.
>
> **As we approach the deadline of rebuttal**, we would like to check if you have any additional feedback or if there are further clarifications we can provide. We truly appreciate the time and effort you’ve invested in reviewing our work.
>
> If you find that our revisions have satisfactorily addressed your concerns, we would be grateful if you could consider reflecting this in your final assessment.
>
> Sincerely,
>
> The Authors

---

### Official Review · Reviewer_zDjf · 2024-11-04

**Soundness:** 2
**Presentation:** 3
**Contribution:** 2
**Rating:** 6
**Confidence:** 3

**Summary:**

This paper studies the quantization problem in Multi-modal LLMs. Specifically, the authors investigate three aspects that lead to performance degradation when applying the straightforward per-tensor static quantization for prefilling multimodal tokens. To address these challenges, this paper presents MQuant with Modality-specific Quantization (MSQ), Attention-Invariant Flexible Switching (AIFS), LayerNorm-to-RMSNorm transformation and Rotation Magnitude Suppression (RMS).

**Strengths:**

1. This paper focuses on a valuable question, i.e. quantization in MLLMs.
2. Well presented with figures and tables.
3. Overall performance is superior to some LLM quantization baselines.

**Weaknesses:**

1. MSQ and AIFS are simply trivial adaptions of per-token dynamic quantization to MLLMs. It's better that this serves as a baseline model.
2. MSQ and MSQ + AIFS exhibit marginal improvement over the per-tensor static baseline in Table 4.
3. Please discuss the overhead of MSQ, otherwise why don't we use token-specific quantization?
4. Although MSQ + AIFS is proposed to address the token increase brought by larger resolution of images, the speedup fails to exhibit great advantages over per-token dynamic baseline with resolution scaling.
5. SliceGPT [1] has already proposed converting LayerNorm to RMSNorm and provides a solution, which you do not mention in the related work. Please discuss the difference between your method in Section 4.2 and the one in SliceGPT.
6. Lack of sufficient technical contribution. Most of the techniques used are from previous work and adapt to MLLM with trivial modifications.
7. Typos. e.g. whthin in line 304 and grammatic errors, e.g. 305 (should be "to show how to transform xxx")

[1] Ashkboos, Saleh, et al. "Slicegpt: Compress large language models by deleting rows and columns." arXiv preprint arXiv:2401.15024 (2024).

**Questions:**

Please see the weakness.

---

> ### Author Response · Authors · 2024-11-21
> **The MSQ and AIFS**
>
> Thank you for reviewing our work and providing useful suggestions. Please check our detailed reply to your questions/comments.
>
> **Q1**: MSQ and AIFS are adaptions of per-token dynamic quantization
>
> **A1**: Please refer to **General Response 1** regarding per-token dynamic and per-tensor static quantization.
> * In fact, MSQ is entirely unrelated to per-token dynamic quantization. It is a **novel static quantization approach specifically designed to address the unique challenges of MLLMs.**
>
> * In MLLMs, there is a significant disparity in the data distribution between visual and textual features. As illustrated in Figure 1.b of the manuscript, the magnitude of visual features is tens to hundreds of times larger than that of textual features. During quantization calibration, this imbalance causes the quantization parameters to be heavily influenced by the large values of the visual features, leading to substantial information loss in the majority of textual features as well as smaller visual features. **MSQ effectively addresses the significant differences in modality distributions, achieving near-lossless precision.**
>
> * However, due to the arbitrary quantity and positioning of different modalities in MLLMs, directly applying MSQ introduces additional and irregular data processing steps, such as slicing, concatenation, and padding. These operations increase memory overhead and reduce the computational efficiency of massive GEMM layers. To address this challenge, we propose **Attention-Invariant Flexible Switching (AIFS) which transforms mixed multimodal tokens into a unified, modality-decoupled, and attention-invariant tensor.** AIFS is performed only once before prefill stage, eliminating the need for dynamic position vectors and preserving computational equivalence throughout the rest of the execution.
>
> In summary, the combination of MSQ and AIFS achieves the same efficiency of per-tensor static quantization while maintaining near-lossless accuracy comparable to the original FP32 model.
>
> **Q2**: The Table 4.
>
> **A2**:   Table 4 aims to present the latency evaluation of the different quantization methods without including the corresponding accuracy. We further provide the detailed latency and accuracy in Table 4.
>
> | Method | Linear Latency (s) | TextVQA Val | DocVQA Val | OCRBench | MME |
> |---|---|---|---|---|---|
> | per-token dynamic | 1.253 (baseline) | 84.32 | 93.61 | 830 | 2269 |
> | per-tensor static | 1.016 (**+23%**) | 40.20 (**-44.12**) | 38.82 (**-54.79**) | 422 (**-408**) | 1082 (**-1187**) |
> | MSQ | 1.085 (**+16%**) | 84.32 | 93.61 | 830 | 2269 |
> | AIFS+MSQ | 1.017 (**+23%**) | 84.32 | 93.61 | 830 | 2269 |
>
> 1. In MLLM quantization, per-tensor static quantization achieves the fastest inference speed (speed upper bound), but it leads to significant performance loss. Although per-token dynamic quantization performs well (accuracy upper bound), the online token-wise computation of scales limits the MLLM's inference speed.
>
> 2. Our proposed MSQ and AIFS aim to achieve the same accuracy as per-token dynamic quantization while reaching the speed of per-tensor static quantization. We have updated Table 4, presenting both speed and accuracy results, and plotted a figure in this anonymous link https://ibb.co/ZB4kKSq. Our MSQ + AIFS achieves speeds nearly on par with per-tensor static quantization while attaining the accuracy of per-token dynamic quantization.
>
> **Q3**: Overhead of MSQ.
>
> **A3**: Please refer to A1 and the results in A2.
>
> **Q4**:  Speedup of MSQ+ AIFS.
>
> **A4**:
> 1. In our original paper, we only presented the acceleration results during the **prefill stage** of MLLM. To provide a more comprehensive comparison, we further report the acceleration results including the **decode stage**. Here, configuration is aligned with Table 4, measuring mean latency (**ms**) of linear layer for decoding 2,000 tokens. A custom kernel was implemented for W4A8 kernel GEMV operations.
>
> | stage | fp16 | Dynamic W4A8 | Ours | Ours+GEMV | Improvement |
> |:---:|:---:|:---:|:---:|:---:|:---:|
> | Prefill | 1690 | 1253 | 1017 | - |**+23%** |
> | decode | 17.5 | 16.4 | 13.06 | 8.2 | **+100%** |
>
> As shown in the table, compared to per-token dynamic quantization, in addition to achieving **23%** speed improvement during the prefill stage, our method achieves an **100% speed up in decode stage**. Overall, our AIFS+MSQ transforms the time-consuming online dynamic quantization into offline static quantization, achieving significant acceleration with almost no loss in accuracy, especially in long sequences.
>
> 2. Notably, in practical applications, using OpenAI's token pricing as an example (https://aigcrank.cn/llmprice), our method can save **~30% in costs**, and this effect is even more pronounced in other MLLMs, as visual tokens are more expensive.
>
> 3. Furthermore, we also think that our kernel has not yet been efficiently optimized for engineering implementation, and further optimizations could yield faster acceleration results.

---

> ### Author Response · Authors · 2024-11-21
> **Difference with SliceGPT**
>
> **Q5**: Difference with SliceGPT
>
> **A5**:  Thank you for your valuable feedback.
>   1. We mentioned SliceGPT in both the related work (**Sec 2 Line 143**) and Method (**Sec 4.2 Line 286**) of the original manuscript and did not overlook this relevant work.
>   2. SliceGPT only designed a **Pre-LN + Rotate scheme** for LLMs and adds a linear layer at the residual connection. Unlike SliceGPT, we further developed a **Post-LN + Rotate scheme** to accommodate the structures commonly found in MLLMs and extended it to better suit MLLMs. Additionally, we incorporate a globally shared rotate matrix, which allows us to remove the additional linear layer at the residual connection and enhance quantization effectiveness without increasing computational overhead. This extension broadens the applicability of the LayerNorm + Rotate approach, making it suitable for both Pre-LN and Post-LN configurations commonly found in various MLLM architectures. The extensive quantization results in Table 2 of the manuscript also demonstrate the effectiveness.
>   3. Particularly, we also presented the **different LayerNorm styles of various MLLM models in Table 7** and discuss the Pre-LN + Rotate Scheme in Appendix.
>   4. We also added more discussion in Method Sec 4.2 to make our contribution clearer. The changes are colored in Blue.
>
> **Q6**: Defend our novelty
>
> **A6**:
> * Our research is rooted in a deep exploration of the unique quantization issues in MLLMs and provides a comprehensive analysis based on these valuable observations, revealing the root causes of performance collapse during MLLM quantization (speed limitation of dynamic per-token, data distribution differences of multi-modal input, sensitive outliers).
>  * To facilitate efficient inference for variable-sequence input tokens, we propose **Modality-specific Quantization (MSQ) and Attention-Invariant Flexible Switching (AIFS)** to support per-tensor static quantization while maintaining lossless accuracy.
>  * To ensure the generalization of our Mquant across various MLLMs, we propose an equivalent transformation from **Post-LN + Rotate scheme**, distinguishing from SliceGPT which only presents pre-LN + Rotate scheme.
>  * We further identified weight outlier magnitudes caused by Hadamard rotation and proposed **Rotation Magnitude Suppression (RMS)** to mitigate it.
>   * Extensive results across five different MLLMs demonstrate the effectiveness and generalizability of our Mquant, which is, to the best of our knowledge, **the first efficient and accurate PTQ solution for MLLMs**.
>   * More importantly, as discussed above, our approach can achieve **objective economic cost savings** in practical deployments and provides **valuable insights for the application of MLLMs on edge devices**.
>
> **Q7**: Typos
>
> **A7**: Thanks! We have fixed it and double checked the grammatical errors.
>
> If there are still any unresolved doubts, please feel free to let us know, and we will make every effort to solve them.

---

> ### Comment · Reviewer_zDjf · 2024-11-27
> **Response to the rebuttal by the authors**
>
> > Q1: Novelty of MSQ and AIFS
>
> Sorry for mistyping the per-token dynamic quantization for per-tensor static quantization. If I understand correctly, the MSQ is per-tensor static quantization adaptive to different modalities, where the scaling factors are different on different modalities, which I believe is a relatively trivial adaption of per-tensor static quantization to MLLMs.
> Although I would credit the AIFS as a technical contribution of this paper, as an efficient implementation of per-tensor static quantization for MLLMs.
>
> > Q2: Marginal performance in table 4
>
> Thanks for your comments and the new table. It makes a lot more sense now. I suggest revising the paper to incorporate the new Table 4, as MSQ + AIFS does not achieve acceleration compared to per-tensor static. Its strength lies in maintaining performance without a decrease in speed.
>
> > Q5: missing related work.
>
> Thanks for the comments. I do recognize that you mention SliceGPT in the related work, yet missing the discussion on LN + Rotate scheme and the solution they propose to convert LN to RMSNorm. This misspecification might lead the readers to believe this work proposes the conversion from LayerNorm to RMSNorm. I would suggest adding a discussion on this in the related work.
>
> In general, I like and encourage the direction this paper works towards, i.e. modality adaption on quantization techniques. However, the proposed techniques in the paper largely build upon previous work. Thus, I can only raise my score to 6.

---

> ### Author Response · Authors · 2024-11-29
> **Thanks for your recognition and encouragement of our paper**
>
> Dear Reviewer zDjf,
>
> Thank you for your constructive comments and valuable suggestions. We greatly appreciate the time and effort you have dedicated to reviewing our manuscript. Your feedback has been instrumental in improving its overall quality and clarity.
>
> **Q1**： Modality-Specific Quantization
>
> **A1**：If you don’t mind, we would like to take a moment of your time to provide more information about **Per-Modality Quantization**.
>
> * While Modality-Specific Quantization involves applying per-tensor static quantization with different scaling factors for different modalities, we want to emphasize that this approach addresses a fundamental and previously unaddressed challenge in the quantization of MLLMs. In MLLMs, the activation distributions of visual and textual tokens differ significantly due to the heterogeneous nature of multimodal data (as shown in Figure 1(b) of our paper). Standard per-tensor static quantization assumes homogeneous distributions and, when applied directly to mixed-modality tokens, leads to severe accuracy degradation because a single scaling factor cannot adequately represent both modalities.
>
> * Per-modality quantization is not a trivial adaptation but a necessary and novel solution tailored to handle these modality-specific distributional discrepancies. Identifying this challenge required thorough theoretical and experimental analysis to uncover the root cause of quantization failures in MLLMs. Much like per-token dynamic quantization in LLMs, which is not simply a trivial adaptation of dynamic quantization, per-modality quantization is an advanced technique specifically tailored for MLLMs, necessitating a re-evaluation of traditional quantization strategies used in such models. **It's not just a trivial extension of static quantization; it demands careful consideration of modality-level variability and alignment, inference framework compatibility, hardware constraints, and the trade-offs between accuracy and performance**. While challenging to implement, it holds promise for improving the performance of quantized MLLMs, especially in resource-constrained environments.
>
> * To be more specific, implementing MSQ involves the interleaved arrangement of visual and textual tokens in MLLMs. Efficiently applying modality-specific scaling factors without incurring additional computational overhead necessitates careful design. This challenge led us to develop the Attention-Invariant Flexible Switching (AIFS) scheme, which reorders tokens into modality-specific sequences while preserving the attention mechanisms.
>
> To highlight the significance of MSQ, we have revised Section 4.1 and Section 1 in the manuscript. We elaborate on the experimental insights that motivated MSQ and explain how it specifically addresses the critical issue of modality-induced quantization errors—a problem not addressed by previous work. In summary, it is the combination of identifying this root problem, proposing **Per-modality Quantization** as a solution, and implementing it efficiently with AIFS that constitutes our contribution.
>
> We hope that this clarification helps convey the importance and novelty of our work. Thank you again for your valuable feedback.
>
> **Q2**： Table 4
>
> **A2**：Thank you for your insightful suggestion. We have revised the manuscript to update the **new Table 4 (Section 5.2)**, which clearly presents the updated results, highlighting that MSQ + AIFS achieves similar acceleration to per-tensor static quantization while maintaining near-lossless accuracy comparable to the Float model. Additionally, we also updated the corresponding description in Section 5.2 to emphasize this point, ensuring that readers understand the significance of maintaining high accuracy without a decrease in speed.
>
> **Q3**： Related Work
>
> **A3**：Thank you for your suggestion. We have added the discussion in Related Work (Section 2.2) and updated the content in the revised manuscript to specifically discuss the main differences between our method and SliceGPT. The updated description is as follows:
>
> > SliceGPT reduces memory demands by designing a Pre-LN + Rotate Scheme for LLMs sparsification based on computational invariance. They achieve this by adding a linear layer in the residual connection (see Appendix A.14). Unlike SliceGPT, we further develop a Post-LN + Rotate scheme to accommodate more vision encoders and extend its applicability to various MLLMs. This enhancement broadens the LayerNorm + Rotate approach, making it suitable for both Pre-LN and Post-LN configurations across various MLLMs.
>
> Sincerely,
>
> The Authors

---

> ### Author Response · Authors · 2024-11-29
> **Further Clarification about novelty**
>
> Thank you for your encouraging words and for appreciating the direction of our research. We would like to clarify and highlight the unique contributions of our work, which have been overlooked or underexplored in existing research.
>
> **1.** Addressing the **inapplicability of existing LLM quantization methods** to MLLMs:
>   * **Unique Challenges in MLLMs**: Through extensive experiments and analysis (see Figure 1 and Tables 2, 5, and 8), we discovered that SOTA quantization methods for LLMs, such as Quarot and GPTQ, do not perform well when directly applied to MLLMs. This is due to the significant distributional differences between visual and textual modalities and the sensitivity of vision encoders to outliers.
>   * **Novel Insight**: Recognizing that existing methods are insufficient for MLLMs is a **critical first step** that highlights the necessity for new solutions tailored to multimodal architectures.
>
> **2.** Development of per-modality quantization:
>   * **Unique Solution for Heterogeneous Data**: MSQ is not a trivial adaptation but a novel approach that applies distinct per-tensor static scaling factors to different modalities within MLLMs. This effectively addresses the substantial distributional discrepancies between visual and textual tokens, a problem not tackled by prior work. As discussed above, much like per-token dynamic quantization in LLMs, per-modality quantization is an advanced technique specifically tailored for MLLMs, necessitating a re-evaluation of traditional quantization strategies used in such models.
>   * **Impact on Accuracy**: As shown in Table 2, MSQ enables us to maintain near full-precision accuracy under challenging quantization settings (e.g., W4A8), which is a significant advancement over existing methods.
>
> **3.** Introduction of Attention-Invariant Flexible Switching (AIFS):
>   * **Efficiency Without Compromising Performance**: AIFS is a critical innovation that allows for the efficient implementation of MSQ by reorganizing tokens into modality-separable sequences while preserving the original attention mechanisms.
>   * **Overcoming Implementation Challenges**: Applying MSQ directly is non-trivial due to interleaved token arrangements in MLLMs. AIFS resolves this, avoiding additional computational overhead and ensuring practicality.
>
> **4.** Theoretical Analysis and Proposal of Rotation Magnitude Suppression (RMS):
>   * **Identifying Limitations of Existing Methods**: In **Section 4.3 and Appendix A.3**, we provide a theoretical analysis showing that online Hadamard rotations, used in methods like Quarot, introduce significant weight outliers in MLLMs, degrading quantization performance.
>   * **Novel Solution**: We proposed RMS, a simple yet effective method to mitigate these outliers. RMS is low-overhead and easy to deploy, significantly improving quantization results in both MLLMs and LLMs, as evidenced in **Tables 5 and 8**.
>
> **5.** Practical Effectiveness and Insights:
>   * **Advancing the Field**: Our work sheds light on previously unexplored issues based on in-depth analysis in quantizing MLLMs, offering insights that can guide future research. Besides, we propose straightforward solutions that enhance the generalizability of our methods. The combination of MSQ, AIFS, Post-LN+Rotate, and RMS leads to substantial gains in quantization performance, enabling us to achieve accuracy levels close to the full-precision models across multiple models and datasets.
> In summary, while our work builds upon concepts from prior research, we introduce significant novel contributions specifically tailored to the unique challenges of MLLMs. By addressing critical gaps and providing effective solutions, we believe our paper advances the state of the art in quantization techniques for multimodal models.
>
> We believe that, after incorporating your suggestions as well as those from the other reviewers, the latest version represents a substantial improvement over the version you evaluated previously (28 Nov 2024). Therefore, we kindly hope that you could take some time to review our improvements and provide a re-evaluation.
>
> Sincerely,
>
> The Authors

---

### Author Response · Authors · 2024-11-21
**General Response1: Background of related quantization techniques**

**In per-tensor static quantization, the quantization parameters (i.e., scale and zero-point) are precomputed for an entire tensor (e.g., weights or activations) and remain fixed throughout inference**. While efficient, this approach often leads to large and unacceptable accuracy loss in MLLMs due to their diverse activation distributions across varying inputs.

**In contrast, per-token dynamic quantization computes quantization parameters on-the-fly for each input token during inference**. This approach incurs significantly higher computational overhead, as the quantization parameters must be recalculated for every input token, along with multiple additional memory traversals.  Such requirements make per-token dynamic quantization unfriendly or impractical for edge devices and some AI accelerators, which struggle with fine-grained dynamic operations [1].  This issue is especially severe in MLLMs, where the token count increases significantly with higher image resolution or more video frames.

**Our MQuant is a novel per-modality quantization approach specifically designed to address the unique challenges of MLLMs quantization**.  MQuant achieves the same efficiency as per-tensor static quantization while maintaining near-lossless accuracy comparable to the original FP32 model.

[1]. MobileQuant: Mobile-friendly Quantization for On-device Language Models, EMNLP 2024

---

### Author Response · Authors · 2024-11-21
**General response and revision update notice**

Dear Reviewers,

We thank you for your  precious reviews. The paper has been revised according to your suggestions. We've carefully examined all the questions and provided answers. Please feel free to discuss with us if any new questions arise. All changes in manuscript are marked with blue.

* Revised the second-to-last paragraph and summarized the main contributions to enhance clarity (suggested by Reviewer FtkM) (Section 1).
* Revised description of Equation (6), and GEMM and W4A8 to eliminate ambiguity (suggested by Reviewer FtkM) (Section 3).
* Added a detailed explanation of the positional embeddings transformations in AIFS. (suggested by Reviewer xqur) (Section 4.1).
* Added differentiation from SliceGPT and made our contribution clearer (suggested by Reviewer zDjf) (Section 4.2).
* Revised the experimental description (suggested by Reviewer xqur) (Section 5.1).
* Added latency measurements in Table 5 for clarity (suggested by Reviewer FtkM) (Section 5.2).
* Added detailed description of position embedding in AIFS and clarified Figure 1 in the Supplementary Materials (suggested by Reviewer xqur).

Sincerely,

The Authors

---

### Author Response · Authors · 2024-11-27
**Sincerely looking forward to your more feedbacks**

Dear Area Chairs and Reviewers,

We sincerely appreciate everyone’s efforts put into the reviewing process, which has significantly contributed to the refinement of our manuscript. We've carefully examined all the constructive questions and have accordingly revised our work. To make the best use of the discussion period and to improve our work, we are eager to know whether our answers well address your concerns, as it is crucial for us to have a candid and thorough discussion to continuously strengthen our method. Please share your thoughts on viewing our reply. We hope to resolve your doubts with our best efforts.

We are ready to respond to any further issues raised. Please let us informed.

Sincerely,

The Authors

---

### Author Response · Authors · 2024-11-29
**General response and further revision update notice**

Dear Reviewers,

We thank you again for your valuable reviews. The paper has been **further revised according to your suggestions**. We've carefully examined all the questions and provided answers. Please feel free to discuss with us if any new questions arise. All changes in the manuscript are marked in blue.

- Revised the discussion about SliceGPT in **Related Work**, specifically discussing the methodological differences (suggested by Reviewer zDjf) (Section 2.2).
- Revised the advantages of MSQ+AIFS and moved them to the corresponding experimental sections to enhance solidity and clarity (suggested by Reviewer FtkM) (Sections 4.1, 5.1, 5.2).
- Revised and streamlined the description of Post-LN+Rotate to highlight our core contribution (suggested by Reviewer FtkM) (Section 4.2).
- Revised the description of RMS to reduce redundant text while adding theoretical support and analysis (suggested by Reviewer FtkM) (Section 4.3).
- Updated Table 4 and added accuracy and speed comparisons for clarity (suggested by Reviewer zDjf) (Section 5.2).
- Added speedup of the prefill and decode stages for MSQ+AIFS in the Appendix to highlight the substantial acceleration effects (suggested by Reviewer zDjf) (Appendix A.2).
- Added algorithm and generalization experiments of RMS on LLMs in the Appendix to demonstrate their effectiveness (suggested by Reviewer FtkM) (Appendices A.7, A.8).
- Added a detailed schematic of the Post-LN+Rotate scheme in the Appendix to highlight the differences with SliceGPT (suggested by Reviewers zDjf and FtkM) (Appendix A.14).

Sincerely,

The Authors

---

### Meta-Review · Area_Chair_26zr · 2024-12-05

**Metareview:**

**Summary of Scientific Claims and Findings**
The paper introduces MQuant, a post-training quantization (PTQ) framework tailored for multimodal large language models (MLLMs). The proposed method addresses challenges such as distributional discrepancies between visual and textual modalities, inference latency due to visual tokens, and performance degradation from visual outlier clipping. The authors propose techniques such as Modality-Specific Quantization (MSQ), Attention-Invariant Flexible Switching (AIFS), LayerNorm-to-RMSNorm transformation, and Rotation Magnitude Suppression (RMS), claiming improvements in both accuracy and speed.

**Strengths**
- The paper tackles an important and underexplored problem of MLLM quantization.
- The methodology is supported by extensive experiments across multiple mainstream MLLMs.
- The authors provided detailed rebuttals and additional experiments to clarify questions raised during the review process.

**Weaknesses**
1. **Limited Novelty**:
   - Several proposed methods, such as the LayerNorm-to-RMSNorm transformation and the use of Hadamard rotation, are adaptations of existing techniques from LLM quantization literature (e.g., SliceGPT and other prior works). The novelty of the contributions is incremental rather than groundbreaking.
   - The core contribution of MSQ and AIFS, while specific to MLLMs, is primarily a direct application of existing concepts like per-tensor static quantization and data reordering, raising concerns about the lack of fundamental innovation.

2. **Lack of Generalization**:
   - While the authors added experiments during rebuttal to address multi-turn and multi-batch inference, these setups are still limited in scope. Broader and more diverse use cases, such as varying batch sizes or more complex sequences in real-world scenarios, are not fully addressed.

3. **Writing and Organization**:
   - Initial drafts of the paper were verbose and poorly structured, devoting excessive focus to prior methods rather than highlighting the novelty of the authors' contributions. Despite revisions, the paper still lacks conciseness and clarity in some sections.

4. **Marginal Improvements**:
   - Performance gains over baselines are relatively minor in key metrics, particularly in Table 4, where speed improvements of MSQ+AIFS are comparable to per-tensor static quantization but fail to showcase clear advantages in practical scenarios.

**Decision Rationale**
While the problem addressed is important and the work demonstrates technical competence, the paper’s contributions are incremental and build heavily on prior work without providing sufficient innovation. The practical significance of the improvements is also limited, with minor gains in performance that do not strongly justify the proposed methods. Moreover, concerns about generalizability and the lack of a significant leap in methodology weigh against acceptance.

**Additional Comments On Reviewer Discussion:**

During the review process, reviewers raised concerns about the paper's novelty, experimental setup, and presentation. The authors addressed these issues during the rebuttal, but the core concerns persisted.

1. **Novelty**: Reviewers questioned the originality of MSQ and AIFS, viewing them as incremental adaptations of existing methods. The authors clarified the challenges specific to MLLMs and highlighted differences from prior work, including SliceGPT. Despite these clarifications, the reviewers found the contributions insufficiently novel.

2. **Experimental Validity**: Reviewers requested additional experiments for multi-batch, multi-turn, and multi-image scenarios and noted marginal performance improvements. The authors provided more results, updated tables, and expanded discussions, but the new evidence did not demonstrate substantial impact or superiority over baselines.

3. **Presentation**: The paper's initial redundancy and lack of clarity were addressed through revisions that condensed text and reorganized sections. While the changes improved readability, they did not alter the perception of limited contributions.

4. **Final Assessment**: Some reviewers raised their scores, acknowledging the authors’ efforts and additional experiments, but skepticism about novelty and practical impact remained. The marginal improvements and incremental nature of the contributions ultimately led to the decision to reject the paper.

---

### Decision · Program_Chairs · 2025-01-22

Reject